# DNA transposons mediate duplications via transposition-independent and -dependent mechanisms in metazoans

Shengjun Tan[1,2,14], Huijing Ma[1,2,3,14], Jinbo Wang[1,2,14], Man Wang[4,14], Mengxia Wang[1,2,3], Haodong Yin[1,3], Yaqiong Zhang[1,2], Xinying Zhang[1,3], Jieyu Shen [1,2,3], Danyang Wang[3,5], Graham L. Banes [6,7], Zhihua Zhang[3,5], Jianmin Wu [4], Xun Huang [3,8], Hua Chen [3,9,10], Siqin Ge[1,3], Chun-Long Chen [11,12✉] & Yong E. Zhang [1,2,3,10,13✉]

Despite long being considered as "junk", transposable elements (TEs) are now accepted as catalysts of evolution. One example is *Mutator*-like elements (MULEs, one type of terminal inverted repeat DNA TEs, or TIR TEs) capturing sequences as Pack-MULEs in plants. However, their origination mechanism remains perplexing, and whether TIR TEs mediate duplication in animals is almost unexplored. Here we identify 370 Pack-TIRs in 100 animal reference genomes and one Pack-TIR (*Ssk-FB4*) family in fly populations. We find that single-copy Pack-TIRs are mostly generated via transposition-independent gap filling, and multicopy Pack-TIRs are likely generated by transposition after replication fork switching. We show that a proportion of Pack-TIRs are transcribed and often form chimeras with hosts. We also find that *Ssk-FB4s* represent a young protein family, as supported by proteomics and signatures of positive selection. Thus, TIR TEs catalyze new gene structures and new genes in animals via both transposition-independent and -dependent mechanisms.

[1] Key Laboratory of Zoological Systematics and Evolution, Institute of Zoology, Chinese Academy of Sciences, Beijing, China. [2] State Key Laboratory of Integrated Management of Pest Insects and Rodents, Institute of Zoology, Chinese Academy of Sciences, Beijing, China. [3] University of Chinese Academy of Sciences, Beijing, China. [4] Key Laboratory of Carcinogenesis and Translational Research (Ministry of Education/Beijing), Center for Cancer Bioinformatics, Peking University Cancer Hospital & Institute, Beijing, China. [5] CAS Key Laboratory of Genome Sciences and Information, Beijing Institute of Genomics, and China National Center for Bioinformation, Chinese Academy of Sciences, Beijing, China. [6] Wisconsin National Primate Research Center, University of Wisconsin–Madison, Madison, WI, USA. [7] CAS Key Laboratory of Computational Biology, Shanghai Institute of Nutrition and Health, Chinese Academy of Sciences, Shanghai, China. [8] State Key Laboratory of Molecular Developmental Biology, Institute of Genetics and Developmental Biology, Chinese Academy of Sciences, Beijing, China. [9] CAS Key Laboratory of Genomics and Precision Medicine, Beijing Institute of Genomics, and China National Center for Bioinformation, Chinese Academy of Sciences, Beijing, China. [10] CAS Center for Excellence in Animal Evolution and Genetics, Chinese Academy of Sciences, Kunming, China. [11] Curie Institute, PSL Research University, CNRS UMR 3244, Paris, France. [12] Sorbonne University, Paris, France. [13] Chinese Institute for Brain Research, Beijing, China. [14]These authors contributed equally: Shengjun Tan, Huijing Ma, Jinbo Wang, Man Wang. ✉email: chunlong.chen@curie.fr; zhangyong@ioz.ac.cn

Transposable elements (TEs), including retrotransposons and DNA transposons, occupy a significant portion of eukaryotic genomes[1]. Although long considered "junk DNA"[2], TEs are now widely accepted as catalysts of genetic innovations by directly contributing to regulatory or coding sequences[3,4] and mediating sequence changes such as duplications or deletions[5,6]. The mechanism responsible for the generation of duplicates affects their evolutionary trajectories[7,8]. Duplicates generated by TEs are more likely to evolve new structures or functions due to the formation of chimeric transcripts or changes in the regulatory context[9–11]. Therefore, the mechanism through which TEs mediate duplications is of broad interest.

Studies, including ours, have shown that long terminal repeats (LTRs) and non-LTR retrotransposons, such as L1 or SVA elements, mediate the retroduplication of host messenger RNAs (mRNAs) in animals and plants[10,12–15]. Among DNA transposons, *Helitrons* duplicate non-TE sequences in animals and plants[16,17], whereas terminal inverted repeat TE (TIR TE)-mediated duplications have often been studied in plants but not in animals[18,19]. Two anecdotal studies in animals show that *P* elements in *Drosophila* capture sequences via two mechanisms upon artificial activation of the transposase. First, *P* elements together with flanking sequences are subject to transposition under transduction (also called the end bypass model, Supplementary Fig. 1a) and thereby accidentally use the downstream sequence as the TIR[20]. Second, *P* elements capture sequences under the gap-filling model[21]. In this model, double-strand breaks (DSBs) occur in two scenarios: (1) the internal sites are broken, possibly induced by secondary structures[22,23] (Supplementary Fig. 1b); or (2) complete TEs are excised due to transposition[24] (Supplementary Fig. 1c). During the repair, the template could switch from the sister strand to adjacent external sequences (called fillers) in 3D proximity, leading to the capture of fillers[21,25]. The whole process in the former scenario (Supplementary Fig. 1b) is transposition-independent.

In contrast, TIR TE-capturing sequences have been extensively studied in plants, particularly rice: one type of TIR TE called *Mutator*-like element (MULE) generates Pack-MULEs (duplicated internal sequences together with flanking MULEs)[26–28]. One-third of Pack-MULEs are multicopy due to several rounds of transposition, as indicated by distinct target site duplications (TSDs, a hallmark of transposition)[26]. The internal sequences are often duplicated in *trans* and derived from interchromosomal sequences[27,29]. Because the source or parental copies are not linked with MULEs or Pack-MULEs, the origination mechanism of Pack-MULEs is incompatible with the end bypass model. The gap-filling model has, therefore, been proposed[27], but whether duplication is associated with transposition remains unknown[18]. Functionally, 40% of Pack-MULEs are transcribed[28] and possibly encode small RNAs or contribute to the 5′ untranslated region (UTR) of host genes[29,30]. Protein-coding Pack-MULEs are rare, and only one protease duplicated by a MULE (*KI-MULE*) has been characterized. However, *KI-MULE* is likely not functional given its repressed expression and heterochromatic location[31].

Here, we consider whether Pack-TIRs (non-TE sequences with flanking TIR TEs) are present in animals, and if so, how they emerge and whether they are functional. To address these questions, we focus on young Pack-TIRs, which more likely retain sequence features indicating their origination mechanism compared with older Pack-TIRs. We identify a conservative dataset that includes 370 Pack-TIRs in 100 animal reference genomes and one Pack-TIR (*Ssk-FB4*, the gene *Ssk* amplified by one TIR TE called *FB4*) family in *D. melanogaster* populations. Sequence analyses of these Pack-TIRs suggest that single-copy Pack-TIRs are mostly generated via a transposition-independent gap-filling process, whereas the birth of multicopy Pack-TIRs is compatible with a new model, which we describe as replication Fork Stalling, Template Switching and Transposition (FoSTeST). Furthermore, we find that an appreciable proportion of Pack-TIRs are transcribed and often chimeric with neighboring genes, and *Ssk-FB4s* may represent one of the youngest functional protein families supported by unique mass spectrometry (MS) peptides and signatures of positive selection. In summary, TIR TEs generate new gene structures and new genes in animals via both transposition-independent and -dependent mechanisms.

## Results

**Hundreds of young Pack-TIRs were identified in animal reference genomes and population resequencing data of *D. melanogaster*.** To generate a comprehensive view of Pack-TIRs during animal evolution, we targeted both reference genomes and population resequencing data ("Methods"). We scanned 100 animal genomes available in the UCSC Genome Browser database[32], which consists of 81 vertebrates (including 57 mammals) and 19 invertebrates. We identified 370 young Pack-TIRs for which both TIR TEs and parental copies of captured sequences could be unambiguously identified. For population analyses, we focused on *D. melanogaster* given the ease of the experiments and the availability of resequencing data generated with the *D. melanogaster* Genetic Reference Panel (DGRP)[33], and we identified one multicopy Pack-TIR family, *Ssk-FB4s*.

**The distribution, copy number, and origination timing of Pack-TIRs in reference genomes suggest a transposition-independent birth process.** To produce an overview of Pack-TIRs harbored by the reference genomes, we analyzed their distribution across species and TE superfamilies. We found that 370 Pack-TIRs were scattered across 55 species with a median number of 4 (Fig. 1a and Supplementary Data 1). The number of Pack-TIRs was correlated with that of consensus TIR TEs in each animal, with approximately two Pack-TIRs per 10,000 TIR TEs (Fig. 1b, $R^2 = 0.60$). Consistent with the literature[34], zebrafish, western clawed frogs, and American alligators were found to be the top three species, with each including more than 120,000 consensus TIR TEs (Supplementary Data 2). These species accordingly encode relatively more Pack-TIRs (8, 25, and 38, respectively). In contrast, probably due to the low content of TIR TEs in birds and insects (median number of 268 and 36, respectively), no Pack-TIR was identified within these species. We found that the distribution of Pack-TIRs across TIR TE superfamilies was analogous to the number of Pack-TIRs predicted based on consensus TIR TEs in each superfamily (Fig. 1c, $R^2 = 0.94$). Most (323 or 87.2%) Pack-TIRs were associated with the top two most common superfamilies, *hAT* (208) and *TcMariner* (115) (Supplementary Data 1), which jointly contributed 81.8% of the TIR TE content in the 100 species (Supplementary Data 2). Although MULEs are active in rice, they are rare (median number of 0) in animals, and no Pack-MULE has been identified in reference genomes.

Because different TE families could exhibit different extents of transposition activity, the linear relationship between the number of Pack-TIRs and consensus TIR TEs appears to suggest a transposition-independent duplication process. More direct evidence came from the inactivity of many consensus TEs[35,36]. In actuality, the consensus sequences of TEs tend to be short, with a median length of only 374 base pairs (bp). Moreover, individual TE copies are different from the consensus by a median divergence value of 18.3% (Supplementary Data 1). These two patterns suggest that TEs likely represent degenerate ancient relics. We thus hypothesized that the majority of Pack-TIRs

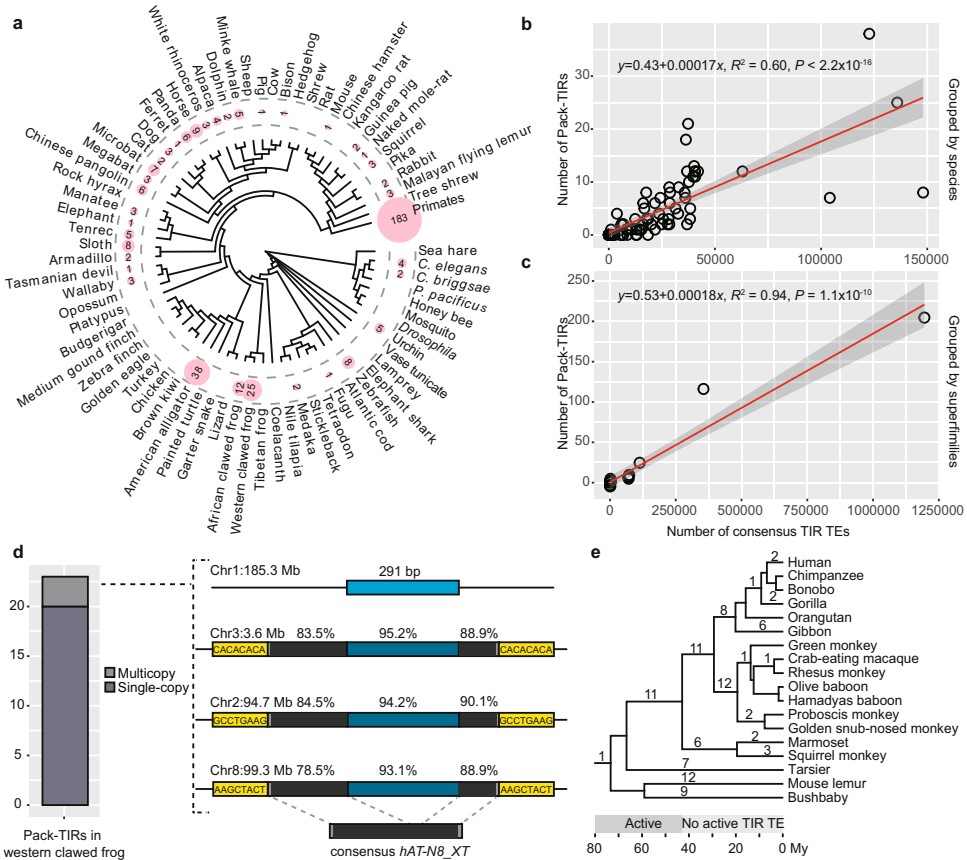

**Fig. 1 Pack-TIRs detected in the animal reference genomes. a** Distribution of Pack-TIRs in 100 animals. The number of Pack-TIRs (if any) is shown close to the taxon name, and the size of the pink background is proportional to the number. Note: the primate order (18 species) and the *Drosophila* genus (11 species) are collapsed, and the total number of Pack-TIRs (if present) is shown. **b, c** Linear relationship between the number of Pack-TIRs and the number of consensus TIR TEs in each species (**b**) or in each TE superfamily (**c**). The shaded areas represent 95% confidence interval. The *F*-test was used to calculate the *P*-value. **d** Pack-TIRs in western clawed frog. The left subpanel shows the number of multicopy and single-copy Pack-TIRs, and the right subpanel shows one case harboring three transposition events with the parental copy (light blue) shown at the top, TIR TE shown at the bottom and the correspondence between TE and Pack-TIR shown as dashed lines. The percentages represent sequence identities relative to the parental copy and the consensus TE. TIRs, non-TIR TE sequences, and internal non-TE sequences are shown in gray, black, and dark blue, respectively. The target site duplications (TSDs) in flanking regions are also marked in yellow. Abbreviations such as "Chr1: 185.3 Mb" indicate a chromosomal coordinate, with Mb representing the number of mega bases. **e** Emergence of Pack-TIRs along the phylogenetic tree of primates. Two periods with active or inactive TIR TEs are marked along the evolutionary time. "My" represents million years.

would be single-copy, due to the absence of a DNA TE mediating replicative transposition. Consistently, we found only four (1.4%) multicopy Pack-TIRs, including three cases in western clawed frogs and one in American alligators (Supplementary Data 1). As exemplified by one frog case, a 291-bp sequence derived from the 5′ untranslated region (UTR) of the *CAPSL* gene was captured by *hAT-N8_XT* and amplified into three different chromosomes (Fig. 1d). Multiple rounds of transposition were supported by three distinct TSDs. However, even for frogs, the proportion of multicopy Pack-TIRs was only 13.0% (3/23, Fig. 1d), which was markedly lower (Fisher's exact test, FET $P = 0.026$) than that of Pack-MULEs observed in rice (36%, 481/1337)[26].

We further tested the hypothesis of a transposition-independent mechanism by examining the origination time of Pack-TIRs along the phylogenetic tree of primates. DNA TEs were transposable in early primate evolution but lost their activity prior to the last ~37 million years (My)[36]. Consequently, if duplication was dependent on transposition, Pack-TIRs should be ancient and shared across multiple primates. By dating and merging orthologous Pack-TIRs across 18 primates, we found 97 unique Pack-TIRs scattered at most branches (Fig. 1e and Supplementary Data 3). As an example, the human genome

harbors 33 Pack-TIRs, including 12 cases directly called by our pipeline (Fig. 1a) and 21 cases called at orthologous loci in nonhuman primates but exhibiting slightly lower alignment quality in humans (not passing the cutoffs, Supplementary Data 3, 4, "Methods"). For these 33 cases, the majority (22 or 66.7%) were generated when DNA TEs were inactive (Fig. 1e), which invalidates the transposition-dependent model.

In summary, the linear relationships among the numbers of Pack-TIRs and consensus TEs, the prevalence of single-copy Pack-TIRs, and the origination timing of Pack-TIRs in primates jointly suggest that these were mainly generated via a transposition-independent mechanism.

**The location and sequence features of single-copy Pack-TIRs indicate a gap-filling process.** We further examined the location and sequence features of Pack-TIRs and found three patterns supporting the gap-filling model. First, the gap-filling model predicts a *cis* duplication bias in which TEs preferentially capture nearby sequences as fillers[21]. Consistently, for 281 unique Pack-TIRs in the reference genomes (after controlling for redundant ones in primates and multicopy cases, Supplementary Data 1), TIR TEs capturing sequences from the same chromosome

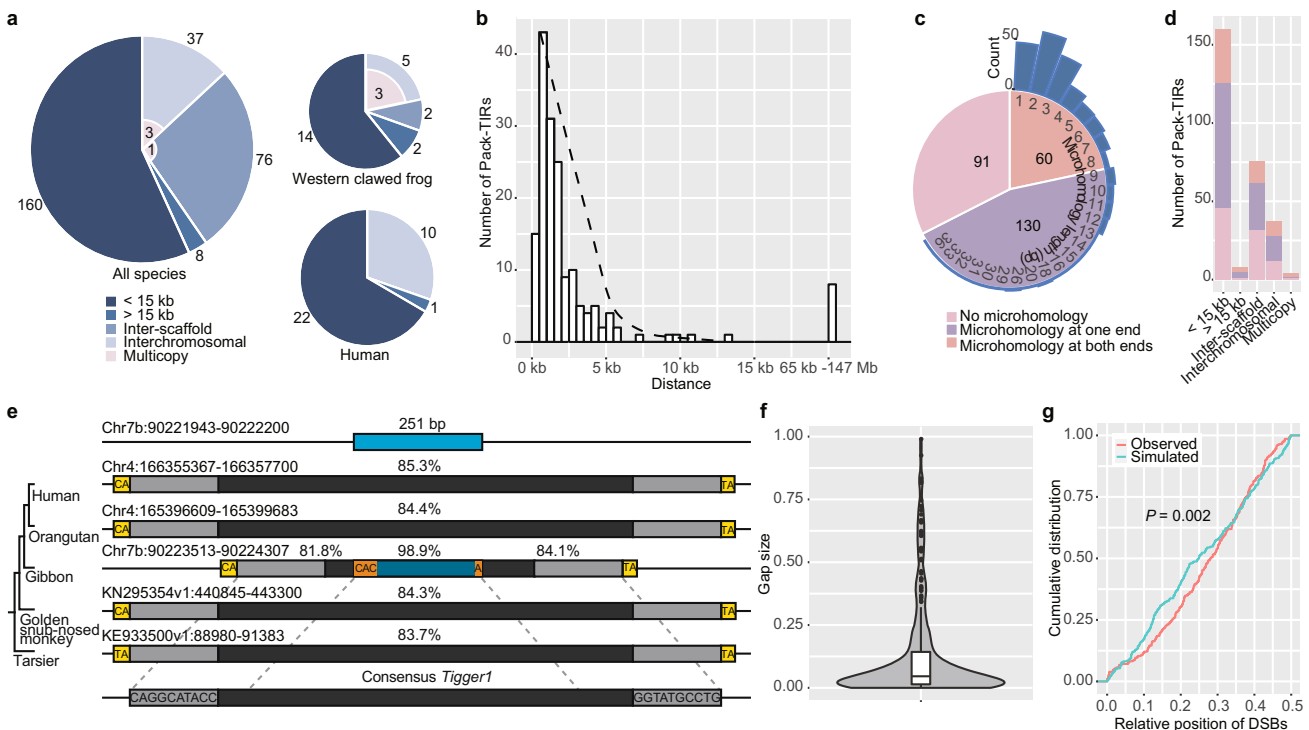

**Fig. 2 Evidence supporting the gap-filling model for the generation of single-copy Pack-TIRs. a** Distribution of Pack-TIRs classified by distance from their parental copies. The pie charts show the statistics for all species, western clawed frogs, and humans, respectively. Numbers of multicopy Pack-TIRs are highlighted in pink. For example, three out of five interchromosomal Pack-TIRs are multicopy in western clawed frogs. **b** Histogram of intrachromosomal Pack-TIRs in terms of distance to their parental copies. The dashed curve shows the lowess fitting result. **c** Distribution of Pack-TIRs classified by the presence of microhomology at two breakpoints. The sidebar shows the histogram of the microhomology length. **d** Distribution of Pack-TIRs with respect to the microhomology presence at breakpoints. Single-copy Pack-TIRs close or far from parental copies and multicopy Pack-TIRs are separately counted. **e** A comparative view of a demo Pack-TIR in gibbons (case #144 in Supplementary Data 1). To simplify, in addition to gibbons, only four phylogenetically representative primates are shown. The figure convention follows Fig. 1d except that the microhomology is marked in orange. Note that the left TSD was mutated from TA to CA in the ancestor of humans and monkeys. **f** Proportion of each gap ($n = 281$ samples) relative to the corresponding consensus TE. The distribution is shown as a violin plot: the box indicates the median (middle line) and the interquartile range (IQR, box limits); outliers are beyond 1.5 times IQR; and the violin curve indicates the probability density of the data. **g** Cumulative density distribution of the observed and simulated relative positions of breakpoints to consensus TEs. Values closer to 0.5 at X-axis indicate positions closer to the middle of the TEs, whereas values close to 0 indicate positions closer to the terminals. The two-sided Kolmogorov-Smirnov test was used to calculate the P-value.

(intrachromosomal duplication) were widespread in animals (59.8%, Fig. 2a). This proportion could be even higher because different scaffolds possibly belong to the same chromosome in species with only draft assemblies. The density of these intrachromosomal Pack-TIRs was negatively correlated with their distance to the parental copies (Fig. 2b), which is similar to *P* element-induced gap repair[21]. Furthermore, 160 (56.9%) Pack-TIRs were situated in the proximity of parental copies (<15 kilobases or kb, Fig. 2a, b), and the distance in the majority of these cases was less than 5 kb. Such a *cis* bias applies for both species currently with active TEs, such as frogs (60.9%), and species without active DNA TEs, such as humans (66.7%, Fig. 2a). In contrast, multicopy Pack-TIRs were always located in distal regions in either frogs (FET *P* = 0.047, Fig. 2a) or alligators (Supplementary Fig. 2). Thus, single-copy Pack-TIRs showed a universal *cis* bias across species with or without transposition activity, which strongly suggested that these might result from a transposition-independent gap-filling process.

Second, template switching in the gap-filling model is expected to occur under the guidance of microsimilarity (a short, similar sequence shared by the two templates; also called microhomology, which was used hereafter) shared by TEs and fillers[25]. Consistently, we found that microhomology (1–36 bp with a median of 2 bp) was present in at least one breakpoint of 190 (67.6%) Pack-TIRs (Fig. 2c). Such a pattern applies for both intra-

and interchromosomal Pack-TIRs and single-copy or multicopy Pack-TIRs (Fig. 2d). We took the breakpoints of 33 human Pack-TIRs as examples to examine whether the microhomology length could be explained by the random association of TEs and fillers ("Methods"). Most (84.4%) did not adhere to the chance effect (*P* < 0.05, Supplementary Fig. 3), which supports the significance of microhomology during template switching.

Third, the gap-filling process induced by the fragile site (Supplementary Fig. 1b) might indicate a longer waiting time between TE insertion and non-TE sequence capture compared with that obtained with the process induced by active transposon excision (Supplementary Fig. 1c). Thus, TIR TEs without duplications should be present at the orthologous locus of outgroup species if duplications are specific to the focal species or lineages. We tested this hypothesis in primates due to their densely sampled phylogeny (Fig. 1e). The majority (91 out of 97, or 94%) fit the expected scenario where orthologous TIR TEs without duplications existed in the outgroup species (Supplementary Data 3). Taking Pack-*Tigger1* (a *TcMariner* element) in gibbons as an example, all four outgroup primates encode orthologous *Tigger1* (Fig. 2e). The analysis of five other Pack-TIRs revealed that no orthologous TEs or TSDs (excision relics) could be detected in related primates (Supplementary Fig. 4a–e). Because these five cases were from sparsely sampled lineages (e.g., tarsier, Fig. 1e), the absence of homologous TEs could be

explained by the lack of closely related outgroups. The last Pack-TIR was shared by primates (Supplementary Fig. 4f), which suggested its ancient origin. Thus, we could not accurately infer its origination process.

Finally, we analyzed whether DSBs induced by breaks within TEs (Supplementary Fig. 1b) or by TE excisions via transposition (Supplementary Fig. 1c) initiated the gap-filling process. We expected that the abortive gap repair in the latter scenario would generate larger deletions with breakpoints biased to the terminals of TEs compared with those obtained in the former scenario. We found evidence compatible with the former scenario: the distances between two breakpoints only account for a median of 4.6% TE sizes (Fig. 2f), and the breakpoints were moderately but significantly skewed toward the internal region of TEs compared with the random uniform distribution (Fig. 2g, "Methods"). The skew is likely due to the enrichment of internal fragile sites.

Thus, the observations of *cis* duplication bias, prevalence of microhomology at breakpoints, post-transposition duplication, and biased distribution of breakpoints toward internal rather than terminal TEs support the notion that a transposition-independent gap-filling process underlies most single-copy Pack-TIRs encoded by animal reference genomes.

**Recurrent transposition generates multicopy Pack-TIRs**. We extended the comparative methodology to four multicopy Pack-TIRs in frogs and alligators (Fig. 2a and Supplementary Fig. 2). Contrary to most single-copy Pack-TIRs in primates, we could not find orthologous TIR TEs for all four Pack-TIRs. Instead, the syntenic regions do not encode Pack-TIRs together with nearby flanking regions. Taking Pack-*hAT-N8_XT* (Fig. 1d) as an example, all three derived Pack-TIRs or *hAT-N8_XT* without duplication were absent in the orthologous loci of the outgroup, the African clawed frogs (Supplementary Fig. 5), which diverged from the western clawed frogs 57 Mya[37]. Thus, either the multi-copy cases do not fit the gap-filling model or the outgroup species is too divergent to provide sequence information indicating the origination process of these Pack-TIRs.

We thus analyzed the multicopy polymorphic *Ssk-FB4s* using the *D. melanogaster* reference genome as the outgroup. Similar to four multicopy Pack-TIRs encoded by frog and alligator reference genomes (Fig. 2a and Supplementary Fig. 2), *Ssk-FB4s* show features of recurrent transposition and long distances between parent and Pack-TIRs. Specifically, the protein-coding gene *Ssk* (*Snakeskin*, essential for intestinal barrier function)[38] encoded in chromosome 3L (coordinate, 20.2 Mb) was almost completely (5′ upstream together with the genic region except ~200-bp 3′ UTR) captured by *FB4*, which was annotated as a *TcMariner* element in Repbase (Fig. 3a). In DGRP lines, we found three *Ssk-FB4* loci, including one on chromosome X (coordinate, 2.7 Mb) and two on chromosome 3R (coordinates, 14.3 Mb and 17.7 Mb, respectively) (Supplementary Data 5). The locus at chr3R: 17.7 Mb was further tandemly duplicated. Because DGRP represents a northern American population, we additionally surveyed the global diversity lines (GDLs) of *D. melanogaster*[39] and confirmed that these populations also harbor only these four copies (Supplementary Data 6). All three loci of *Ssk-FB4* share the same chimeric structure but distinct TSDs (TACATATATG at chr3R: 14.3 Mb, AAATTAAAC at chr3R: 17.7 Mb, and CATGTAGCG at chrX: 2.7 Mb), which suggests recurrent transpositions. Consistently, *FB4* is known as a nonautonomous element transposed by an unknown transposase[40,41]. Notably, since *FB4* has long TSDs (9 to 10-bp) and TIRs (~700 bp), it more likely belongs to MULE rather than *TcMariner* superfamily, which generally have short

TSDs (2-bp TA) and TIRs (<100 bp)[5]. Thus, *Ssk-FB4s* represent Pack-MULEs in *Drosophila*.

To examine the origination process of *Ssk-FB4s*, we determined which might be the first *Ssk-FB4* to initially emerge and which *FB4* element might be involved. By split-read-based genotyping ("Methods"), we found that the X-linked copy was present in relatively more lines (24.2% *vs.* 6.8–16.5%, Fig. 3b), which suggests its earlier origin. Consistently, by reconstructing the phylogenetic tree of the four copies with *Ssk* as the outgroup, we found that the X-linked copy was the most similar to the parent, as shown by the shortest branch length (Fig. 3c). Thus, we inferred that X-linked *Ssk-FB4* is the founding member of the whole multicopy family. If the aforementioned transposition-independent gap-filling model works, an *FB4* element would be present around chrX: 2.7 Mb in fly populations. However, we did not find *FB4* insertion or TSD (excision relic) in the fly reference genome or populations including DGRP and GDL, which suggested that *FB4* is not situated in this locus. We, therefore, inferred that *Ssk-FB4* did not fit the gap-filling model. Among the six full-length *FB4s* present in the reference genome, we deduced that the one at chr3L: 20.8 Mb likely mediated the emergence of *Ssk-FB4* given its highest sequence similarity relative to *Ssk-FB4* (Supplementary Fig. 6) and the high population frequency (present in five out of six sampled lines, Supplementary Data 7).

Motivated by the concurrence of replication and transposition[42] and a replicative mechanism shaping structural variations, i.e., replication Fork Stalling and Template Switching (FoSTeS)[43], we hypothesized that a new model, named replication Fork Stalling, Template Switching and Transposition (FoSTeST), drove the recombination of *Ssk* at chr3L: 20.2 Mb and *FB4* at chr3L: 20.8 Mb and the subsequent shuffling to chrX: 2.7 Mb. We found two lines of evidence supporting this model. Specifically, the TIR of *FB4* is repetitive and associated with frequent rearrangements[44,45]. Consistently, we found that all six *FB4s* harbor structural variations (e.g., tandem duplications, Supplementary Data 7). Thus, replication fork stalling and DSBs occur in these repetitive regions, which leads to frequent template switching. Furthermore, similar to the *cis* duplication bias in the gap-filling model (Fig. 2a, b), *Ssk* and *FB4* (chr3L: 20.8 Mb) are spatially close due to high-order DNA folding, which is supported by DNA interaction data (Fig. 3d and Supplementary Table 1, "Methods"). A 2-bp microhomology (AA) at one breakpoint further facilitates the switching, although a 45-bp de novo insertion was situated at the other breakpoint (Supplementary Fig. 7). A transposon captured this newborn *Ssk-FB4* and moved it to chrX, whereas the 3L-linked *FB4* was repaired because *Ssk-FB4* could not be detected in this locus within fly populations. Subsequently, the X-linked *Ssk-FB4* was subjected to further transpositions.

Taken together, the absences of orthologous TEs without duplication across multicopy Pack-TIRs in frogs, alligators, and flies suggest that multicopy Pack-TIRs were likely generated through the FoSTeST process.

**An appreciable proportion of Pack-TIRs are transcribed and the majority of these are expressed as chimeric RNAs with flanking sequences**. We then inferred whether Pack-TIRs are potentially functional by analyzing their sequence and expression features. First, the size of the duplicated regions was small, with a median size of 224 bp (Fig. 4a), which was similar to that of Pack-MULEs (305 bp)[26]. The size of multicopy Pack-TIRs appeared to be larger, although this difference was not statistically significant (Fig. 4a). Despite their small size, 13.4% of Pack-TIRs captured exonic sequences, and this value was 3.2-fold higher than the

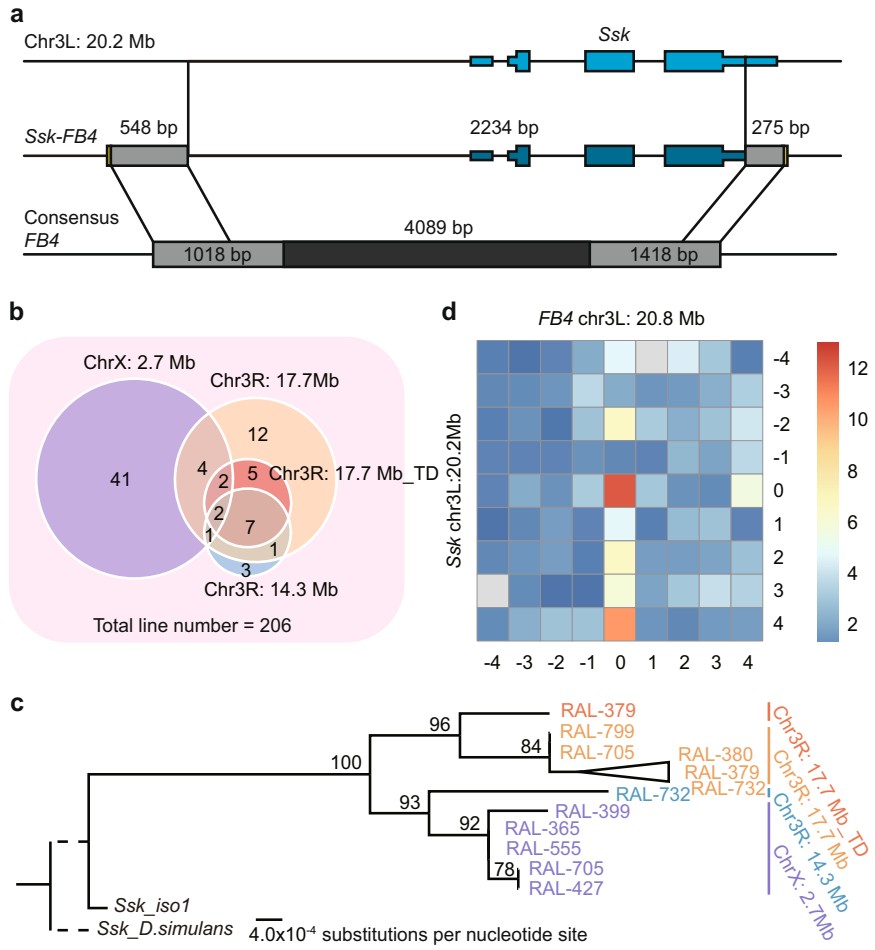

**Fig. 3 Origination and spread of *Ssk-FB4s* in *Drosophila* populations. a** Schematic show of the *Ssk-FB4* gene structure. The figure convention follows that of Fig. 1d with the exception that the thinner boxes, the thicker boxes, and the intervening lines represent the UTR, coding exons, and introns of *Ssk*, respectively. **b** Venn diagram showing the numbers of lines harboring different S*sk-FB4* copies. "TD" represents the tandemly duplicated copy at chr3R: 17.7 Mb. Note: some lines contain more than one copy. **c** Phylogenetic tree of *Ssk-FB4s* and *Ssk*. The genomic sequences of *Ssk* in the reference strain of *D. melanogaster* and the closely related species *D. simulans* were used as the outgroups. *Ssk-FB4s* sequenced in a set of *Drosophila* lines served as ingroups with the colored IDs referring to individual DGRP lines (e.g., RAL-379). Branches with bootstrap scores lower than 60 were collapsed. The ancestral branch of *Ssk-FB4s* and *Ssk* and the branch leading to *D. simulans* are shown as dashed lines because the exact length could not be polarized without outgroups. Four copies of *Ssk-FB4* were color-coded. **d** Chromosome conformation capture (Hi-C) data of *Ssk* (*Y*-axis) and *FB4* at chr3L: 20.8 Mb (*X*-axis). The flanking regions of both loci are shown such that each bin represents a 10-kb window and bin 0 indicates *Ssk* and *FB4*. In total, a 90-kb region was covered. The normalized interaction strength is color-coded.

background proportion (FET *P* < 0.001, Supplementary Fig. 8). This excess could be underestimated because most animal genomes are likely underannotated. Consistently, we found a stronger pattern (21.2% or 4.5-fold, Fig. 4b) in the human genome, which was better annotated. Disproportionate exonic duplications suggest that they are more likely expressed and maintained by natural selection. By analyzing a transcriptome dataset covering four tissues, we found a consistent pattern: five out of eight (62.5%, Fig. 4b) exonic duplications were transcribed, as supported by at least five unique mapping reads ("Methods"), which was significantly higher (FET *P* = 0.036) than intronic and intergenic duplications (5/18 and 0/7, respectively).

We reconstructed the gene structure of 10 transcribed Pack-TIRs in humans via targeted de novo transcriptome assembly of strand-specific reads ("Methods") and found that the captured sequences were always cotranscribed with at least one flanking TE and that the majority (8) were further fused with adjacent genes. Specifically, two Pack-TIRs with exonic sources and four with

intronic sources serve as part of retained introns (Fig. 4c and Supplementary Fig. 9a). This pattern is consistent with the prevalence (75% of genes) of intron retention in mammals, which regulates transcription and splicing[47]. A second scenario is 3′ UTR elongation: a coding exon together with part of neighboring introns of *ERP44* was coopted as part of the 3′ UTR of general transcription factor IIH subunit 5 (*GTF2H5*, Fig. 4d), and one intronic region of H2B clustered histone 15 (*H2BC15*) was duplicated and coopted as part of its own 3′ UTR. Considering the essentiality of *GTF2H5*[48] and *H2BC15*[49], modification of their 3′ UTRs might have a functional consequence. Note: the transcription of these eight cases is likely driven by promoters of host genes because their transcriptional orientation is the same as that of the host genes in seven cases and their expression breadth is identical to that of the host genes in six cases (e.g., *GANQ-MER5A*, Fig. 4c). The last two cases are relatively long: a retrogene *ARF6*[50] was almost completely captured into the *MER20* transposon, but this duplication could also reflect a

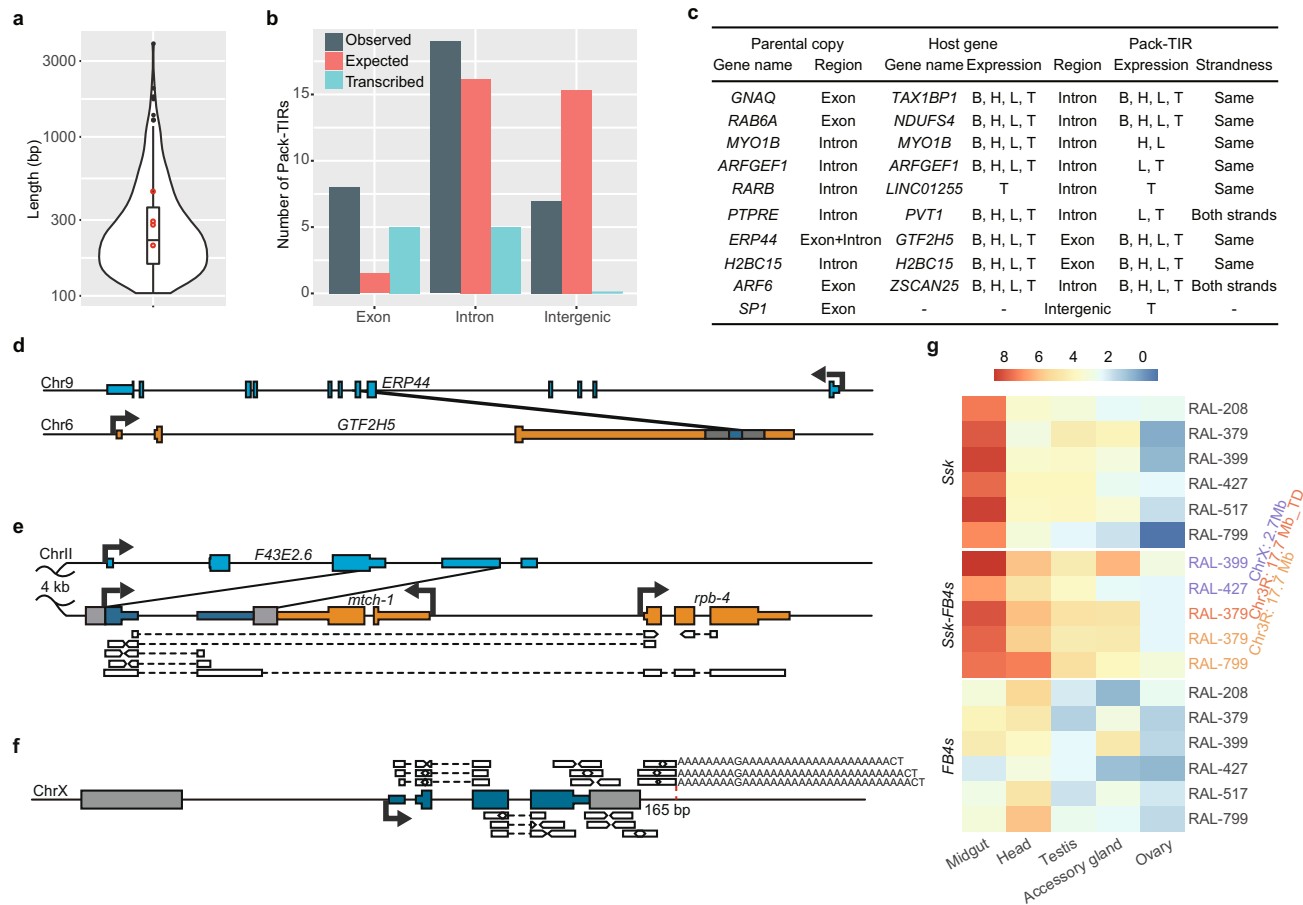

**Fig. 4 Sequence and transcriptional features of Pack-TIRs. a** Length distribution of the internal sequences ($n = 281$ samples). The distribution is shown as a violin plot in Fig. 2f. The four red dots represent the multicopy Pack-TIRs. **b** Distribution of parental sequence types in humans. "Observed" shows the actual numbers of Pack-TIRs divided by the functional types of their parental sequences; "Expected" shows the numbers calculated based on the proportion of all annotated exons and introns; and "Transcribed" shows the numbers of transcribed Pack-TIRs in each category. **c** Transcribed human Pack-TIRs. Host genes indicate the genes into which Pack-TIRs were inserted. "B", "H", "L", and "T" indicate the brain, heart, liver, and testis, respectively. **d** One human example showing 3′ UTR chimerism. The figure convention follows that of Fig. 1d with the exception that the host gene is marked in orange and the transcription direction is shown by thick arrows. **e** One worm example showing 5′ UTR chimerism. The parental gene is close to the host gene at a 4-kb distance. The boxes connected by dashed lines indicate the spliced RNA-sequencing (RNA-seq) reads. Blank arrows indicate paired-end Illumina short reads. The bottom long read was generated on the Nanopore platform ("Methods"). Note: *mtch-1* and *rpb-4* are divergently transcribed. **f** Gene structure of *Ssk-FB4* in *D. melanogaster* at chrX: 2.7 Mb. *Ssk-FB4* is highly transcribed in the midgut, and only a few representative spliced reads spanning breakpoints between the internal sequence and TE and reads supporting continuous transcription toward 165-bp (marked by a red dashed line) downstream regions are shown. Note: the reads containing poly(A) tails have non-A residues, consistent with a previous report[46]. **g** Heatmap showing the expression profile of *Ssk, Ssk-FB4s,* and *FB4s* in five tissues in *D. melanogaster*. The expression levels were quantified as log$_2$ transformed TPM (transcripts per million mapped reads, $n = 2$ biologically independent replicates).

recurrent retroposition event (Supplementary Fig. 9b, "Methods"); the 3′ UTR of *SP1* was duplicated with *MER20* as a noncoding RNA (ncRNA, Supplementary Fig. 9c). In both cases, their expression appears shaped by flanking host regions and/or *MER20* transposons with known promoter activity[51].

We extended analogous analyses to model invertebrates, including worms and flies, and found similar chimerism between Pack-TIRs and neighboring sequences. For four Pack-TIRs encoded by the worm reference genome, we analyzed a whole-body RNA-seq dataset and found that two were expressed as chimeric 5′ UTR and ncRNA, respectively. The former case is worth noting: part of the 3′ UTR and the terminal coding exon of *F43E2.6* were copied to the upstream of the nearby essential gene, i.e., the RNA polymerase II subunit (*rpb-4*). Short- and long-read RNA-seq data revealed two isoforms, and Pack-TIRs provided alternative 5′ UTRs (Fig. 4e). Thus, similar to *GTF2H5* and *H2BC15*, the gene structure of *rpb-4* was changed, which could

have a functional consequence. Note: because this Pack-TIR was inserted into the upstream region of *rpb-4* with the intervening gene *mtch-1* on the antisense strand, its transcription appears driven by the 5′ flanking transposon.

The polymorphic *Ssk-FB4s* in flies are expected to be expressed under the joint regulatory context of *Ssk* and *FB4* (Fig. 3a). To test this hypothesis, we analyzed RNA-seq data ("Methods") by taking advantage of the unique nucleotides of *Ssk* and individual *Ssk-FB4*. We first reconstructed the gene structure of the copy with the highest frequency, i.e., X-linked *Ssk-FB4* (Fig. 3b). We found numerous reads compatible with the original gene structure of *Ssk* and poly(A) tail-containing reads, which indicated a new transcription termination site 165 bp downstream of *Ssk-FB4* (Fig. 4f). Thus, due to loss of the 3′ terminus, X-linked *Ssk-FB4* was fused with the adjacent sequence. Analogously, with the exception of *Ssk-FB4* at chr3R: 14.3 Mb, for which we lacked the corresponding DGRP lines, we identified reads supporting the

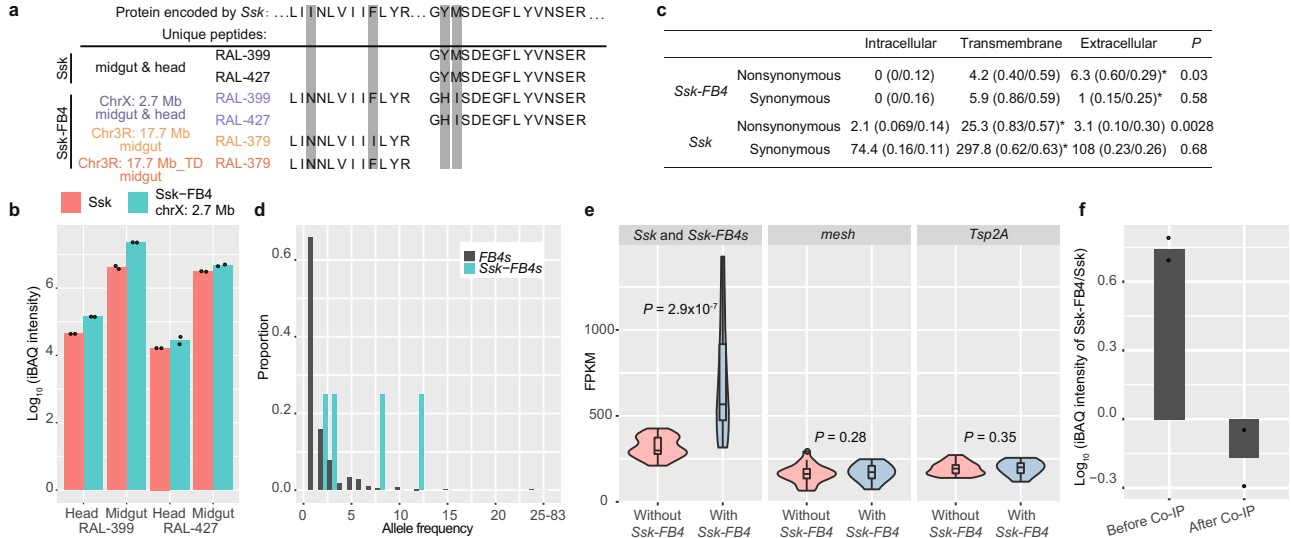

**Fig. 5 Ssk-FB4s represent a young rapidly evolving protein family. a** Unique peptides of Ssk and Ssk-FB4s. Three Ssk-FB4s are color-coded as in Fig. 3c. The top row shows the sequence of Ssk, whereas the distinct residues between Ssk and individual Ssk-FB4s are marked in gray. **b** Abundance (intensity-based absolute quantification, iBAQ) of Ssk and X-linked Ssk-FB4 in the RAL-399 line. Dots indicate 2 technically independent replicates. **c** Numbers of nonsynonymous and synonymous changes across three functional regions in Ssk-FB4s and Ssk. The upper polymorphism data are based on the codon-level alignment of Ssk-FB4s (Supplementary Fig. 14a). The lower divergence data of Ssk are based on the whole *Drosophila* genus (Supplementary Fig. 14e). Each cell is shown as the "number of changed sites (observed/expected proportion of changed sites)". *P-values were calculated for the regions of interest with the one-sided proportion test. **d** Allele frequency spectrum of FB4s and Ssk-FB4s in 83 DGRP lines. **e** Transcription of Ssk, mesh, and Tsp2A in the gut across lines with ($n = 15$ samples) or without Ssk-FB4s ($n = 23$ samples). The distribution is shown as violin plots in Fig. 2f. Because the expression profiles of Ssk and Ssk-FB4s are largely similar, we simply merged the expression of Ssk and Ssk-FB4s. The one-sided Wilcoxon rank-sum test was used to calculate the P-value. **f** Relative abundance of Ssk and Ssk-FB4 in RAL-399 cells before and after co-immunoprecipitation with an antibody against Mesh. Dots indicate two biologically independent replicates.

Ssk-like gene structure for Ssk-FB4 at chr3R: 17.7 Mb and a tandemly duplicated locus (Supplementary Fig. 10). Based on Ssk-derived regions, we quantified the expression of Ssk and three Ssk-FB4s across five representative tissues and identified four patterns (Fig. 4g): (1) Ssk-FB4s are similarly expressed across lines and across copies (Pearson $r^2 = 0.93$, $P = 0.023$), suggesting analogous transcriptional control; (2) similar to Ssk[52], Ssk-FB4s are predominantly expressed in the midgut with a median expression reaching 120, which is higher than 95.0% of genes (Supplementary Fig. 11); (3) in the midgut, Ssk-FB4s sometimes have an even higher expression level than Ssk (457.9 vs. 358.8 in line RAL-399); and (4) Ssk-FB4s are upregulated in the head compared with Ssk (71.5 vs. 13.9, Wilcoxon rank-sum test $P = 0.002$, Fig. 4g) possibly due to the regulatory context of FB4s driving biased expression in the head. Thus, different from repressed KI-MULE[31], Ssk-FB4s are strongly transcribed.

Overall, the profiling of limited transcriptome diversity across three species revealed that 30% or more Pack-TIRs are transcribed as chimeric transcripts. Whether this is generalizable across all animal Pack-TIRs warrants further analysis.

**Ssk-FB4s likely represent a functional protein family evolving under positive selection.** We finally examined whether Ssk-FB4s can encode proteins. We collected protein bands with molecular weights in the range of 15–20 kDa (the mass of Ssk or Ssk-FB4 is ~17 kDa) and used a mass spectrometer to search for peptides ("Methods"). We identified high-quality unique peptides encoded by three Ssk-FB4 copies in available fly lines (Fig. 5a and Supplementary Fig. 12, and Supplementary Table 2). We quantified the expression[53] of X-linked Ssk-FB4, which exhibited the highest frequency, and found that its protein level largely mirrors its transcriptional pattern (Fig. 4g and 5b): (1) Ssk-FB4 is

upregulated in the midgut relative to its expression in the head; (2) the expression intensity of Ssk-FB4 is 5.5-fold higher than that of Ssk in the RAL-399 line midgut; and (3) Ssk-FB4 exhibited higher intensity than Ssk in the head of both the RAL-399 and RAL-427 lines.

Because translation does not necessarily mean function affecting fitness[54], we analyzed whether Ssk-FB4s were subject to positive selection by examining the substitution patterns and frequency distribution. First, because Ssk encodes a membrane protein[38], substitutions at approximately neutral sites, i.e., synonymous sites, would evenly accumulate[55,56] in intracellular, transmembrane, and extracellular regions (Supplementary Fig. 13a). Consistently, there is no enrichment of synonymous substitutions in three regions of Ssk or Ssk-FB4s (Fig. 5c). In contrast, substitutions at functional sites, i.e., nonsynonymous sites, were overrepresented in the transmembrane domains of Ssk and the extracellular regions of Ssk-FB4s, respectively (Fig. 5c and Supplementary Fig. 14b, c). Given the even distribution of synonymous substitutions, these overrepresentations suggest weaker purifying selection or stronger positive selection in the corresponding functional regions. Second, polymorphic FB4s show low allele frequencies, with 65.8% as singletons (Fig. 5d, "Methods"). This phenomenon suggests that the spread of FB4s is repressed by negative selection especially considering the generally deleterious mutagenic nature of TEs[57] (Supplementary Fig. 13b). In comparison, the four Ssk-FB4s have a higher frequency than FB4s (Fig. 5d). This contrast was confirmed by independent frequency data of FB4s (Supplementary Fig. 15a). The increase in the Ssk-FB4 frequency is either because these are less deleterious than FB4s or because they are subject to positive selection. We performed a test by searching for the signal of selective sweep[58], where haplotypes harboring focal mutations would have no time to accumulate many mutations if positive

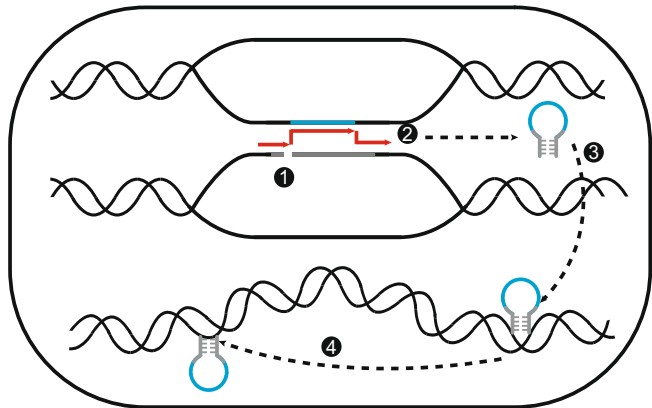

**Fig. 6 Working model of FoSTeST.** The model includes four steps: (1) replication fork stalling; (2) template switching; (3) transposition; and (4) potential amplification via a new round of transposition.

selection drives the rapid increase of these haplotypes (Supplementary Fig. 13c). For two high-frequency copies applicable for this analysis, we indeed found the depletion of linked single nucleotide polymorphisms (SNPs, Supplementary Fig. 15b, c) that could not be explained by chance ($P < 0.05$, Supplementary Fig. 15d, e).

Ssk mainly exerts its function by forming a protein complex with Mesh and Tsp2A at smooth septate junctions (sSJs)[52,59]. Possibly due to the stoichiometric balance between protein complex members[60], these sSJ proteins appear to be subject to coregulation and thus maintain a dosage balance. For example, in response to environmental changes, *Ssk* and *mesh* are consistently up- or downregulated[61]. Therefore, we analyzed whether *Ssk-FB4s* are coregulated with *mesh* or *Tsp2A*. We first found that the expression of *mesh* or *Tsp2A* was not upregulated despite the extra dosage of *Ssk-FB4s* (Fig. 5e and Supplementary Fig. 16). Moreover, after co-immunoprecipitation with an antibody against Mesh, we quantified the protein intensity of Ssk-FB4 and Ssk. Despite the 500% abundance of X-linked Ssk-FB4 relative to that of Ssk in the fly midgut, the former abundance dropped to 70% of the latter in the immunoprecipitates (Fig. 5f). Thus, Ssk-FB4s appear to not be involved in the protein complex of Ssk/Mesh/Tsp2A or only weakly interact with this complex. Together with the contrastive substitution pattern (Fig. 5c), *Ssk-FB4s* and *Ssk* very likely have different functions.

Therefore, *Ssk-FB4s* might have rapidly acquired a novel functional role by modifying extracellular regions under the action of positive selection.

## Discussion

Our study identified hundreds of Pack-TIRs in animals and provided insights into the mechanism underlying their emergence and how they drove functional evolution.

Specifically, our results, particularly the findings that the emergence of Pack-TIRs when DNA TEs were inactive in primates (Fig. 1e and Supplementary Data 3) and that the distribution of breakpoints was biased to internal regions (Fig. 2g), indicate that the transposition-independent gap-filling model (Supplementary Fig. 1b) underlies the formation of most single-copy Pack-TIRs. Certainly, the transposition-dependent gap-filling model (Supplementary Fig. 1c) might also generate a small proportion of single-copy Pack-TIRs. Their copy numbers did not increase, possibly due to low activity of the corresponding TEs or negative selection[62]. Our analyses also hint toward a new FoSTeST mechanism generating multicopy Pack-TIRs. The working model of FoSTeST is as follows (Fig. 6): (1) during DNA

replication, the fork is stalled in TIR TE, and DSB occurs; (2) spatial proximity facilitates template switching, leading to a chimeric Pack-TIR; (3) Pack-TIR is immediately excised by a transposase and moved to another locus, whereas the broken TIR TE is repaired using the sister strand; and (4) the transposase potentially drives further amplification of the Pack-TIR. Although steps 1 and 2 also occur in the gap-filling process (Supplementary Fig. 1b), the newborn sequence is only moved in step 3 of FoSTeST. In both the gap-filling and FoSTeST models, template switching causes duplications. Similar processes have been widely reported in human genetics and cancer genomics[43,63,64], and Pack-TIRs could emerge analogously. Because the gap-filling model has been proven with *P* element in *Drosophila*[21], a similar *P* overexpression system could be used to test the FoSTeST model in the future.

Whether the gap-filling or FoSTeST model also applies to the formation of Pack-MULEs in plants is worth exploring, particularly considering that the comparative framework used in this study is becoming applicable to rice given the recent accumulation of rice genomic data[65,66]. Compared with animals, the FoSTeST model is likely more important in plants, where TIR TEs are more active (e.g., MULEs in rice) and more interchromosomal and multicopy Pack-MULEs exist[26,67]. Moreover, similar to Pack-MULEs[27], multicopy Pack-TIRs also tend to capture high GC-content regions, although single-copy Pack-TIRs do not show this pattern (Supplementary Fig. 17).

With respect to the evolution of new gene structures and new genes in animals, TIR TEs are underappreciated. It could be argued that the rate of TIR TE-mediated duplication was low (Fig. 1a). However, because we used multiple criteria to identify young cases associated with parental sources and flanking TIR TEs, our list of Pack-TIRs is highly conservative ("Methods"). Specifically, our survey identified only 12 Pack-TIRs in humans (Fig. 1a). However, manual curation showed that we missed 21 cases identified in other primates (Fig. 1e), mainly because orthologous cases in humans are slightly below the cutoffs (e.g., identity cutoff with parental copies). Moreover, when we adopted relaxed criteria by removing the requirement of the presence of a parental copy, we identified 199 (including 12 in the original scan) Pack-TIRs in humans. Third, we required the presence of TEs on both sides, whereas they rapidly degenerated via subsequent mutations. Taking one multicopy Pack-TIR in frogs as an example, three out of five copies already lost or gained a TE on one side despite their young ages, as suggested by the high identity detected (Supplementary Fig. 18).

Similar to other duplication mechanisms[9], TIR TE-mediated mechanisms, including both the gap-filling and FoSTeST models, affect the functional trajectories of duplicates. Such a predisposing effect could be demonstrated at both the mRNA and protein levels. Specifically, due to the fragmented nature of Pack-TIRs, they mainly fine-tune the structure of host genes at the insertion site. Pack-TIRs expand the 5′ modification exerted by Pack-MULEs[29] by adding intron retention and 3′ UTR elongation. Certainly, Pack-TIRs could be transcribed as antisense transcripts for the parental genes, which occur in two bidirectional cases derived from *PTPRE* and *ARF6* in humans (Fig. 4c). Thus, regulation as small noncoding RNAs due to sense/antisense pairing, as observed with Pack-MULEs[30], could also occur for Pack-TIRs. Because *KI-MULE* appears to be unexpressed[31], *Ssk-FB4s* are the first identified potentially functional coding Pack-TIRs (Fig. 5), and their evolution appears facilitated by the FoSTeST model. With increases in the copy number, they likely evolve under the adaptive radiation model or innovation, amplification, and divergence (IAD) model[7,68]. In other words, *Ssk* exhibits trace-level beneficial side activity (innovation) but cannot be optimized due to the antagonistic constraint of its main function. With

amplification to increase the mutational targets, positive selection specifically targets the side function of the derived copies and leads to divergence. Consistently, substitutions mainly occur in transmembrane (improving the main function) and extracellular (improving the side function) regions of *Ssk* and *Ssk-FB4s*, respectively; and Ssk-FB4s weakly interact with Mesh or Tsp2A. Furthermore, *Ssk* does not accumulate any nonsynonymous SNPs or coding indels but three synonymous SNPs in the DGRP population. In contrast, *Ssk-FB4s* harbor an excess of amino acid changes (10 nonsynonymous SNPs and two in-frame coding indels *vs.* five synonymous SNPs, FET $P = 0.049$; Supplementary Fig. 14a and 14d).

The side function of *Ssk* or the major function of *Ssk-FB4s* is intriguing. Although *Ssk* is known to be involved in the intestinal epithelial barrier, gut homeostasis and immune response[59,69,70], no studies have pinpointed the key residues underlying these processes; Therefore, we could not infer the function of Ssk-FB4s based on amino acid differences relative to Ssk. Nonetheless, *Ssk-FB4s* are likely involved in similar processes as *Ssk* given their similar expressional control. To infer the functionality of *Ssk-FB4s*, we analyzed public genome-wide association study (GWAS) datasets ("Methods") from flies subjected to microbial infection and their gut homeostasis or immune response was challenged. We found that the presence of *Ssk-FB4s* was always associated with stronger resistance against viral, bacterial and fungal infection in all datasets[71–73]: the comparison reaches marginal significance in one set with female flies subjected to bacterial infection ($P = 0.06$, Supplementary Fig. 19), and significance in another two sets with flies subjected to fungal infection ($P = 0.002$ and $P = 0.02$, respectively). Whether and how they shape these phenotypes warrant further studies.

We noted that studies of new genes mainly focused on fixed genes[74–76]. The youngest or most polymorphic new genes have rarely been studied, with the exception of a few tandemly duplicated or retroduplicated cases, such as *amylase*, *Sdic* or *retro-FGF4*[77–79]. In this respect, the *Ssk-FB4* family represents the only reported case mediated by DNA transposons. Despite their young age, their possible function suggested by proteomic, evolutionary, and association analyses adds to the expanding picture in which new genes could rapidly gain functionality[80,81].

Finally, our results together with those obtained in prior studies[10,12,14,15] show that both DNA transposons and retrotransposons catalyze duplications in animals. Despite their different transposition mechanisms and distribution across species, DNA transposons and retrotransposons share four features. First, the end bypass model (Supplementary Fig. 1a) can apply to the 3′ transduction mediated by TIR TEs[20], *Helitrons*[82], L1s[15,83], and SVAs[14]. Second, template switching during the gap-filling (Supplementary Fig. 1b, c) or FoSTeST model (Fig. 6) is also commonly observed in the duplication process mediated by *Helitrons*[84], L1s[85] and LTR retrotransposons[10]. Third, duplicates flanked by TEs could act as pseudo TEs and amplify via further transpositions, such as the *Ssk-FB4* family described here, the *AMAC* family mediated by SVA[14,86] and the *CG17604_r* family mediated by LTR retrotransposon[10]. Fourth, chimerism between duplications, flanking TEs or insertion sites is widespread not only for Pack-TIRs but also for *Helitrons*[16], LTRs[10], L1s[85,87], and SVAs[14]. All of these features confer TEs with a strong capability of shuffling genetic materials and endorse TEs as a vibrant force in shaping the evolution of new genes and new gene structures in animals.

## Methods

### Identification and analyses of Pack-TIRs in 100 animal reference genomes.
We identified young Pack-TIRs (Supplementary Fig. 20a) with unambiguous parental copies by modifying a previous method[67]. Specifically, the UCSC Genome Browser database hosts 106 animal species (https://genome.ucsc.edu, December 2016), in which TEs have been annotated in 100 genomes. After downloading the genome sequences and TE annotations for these species, we searched for sequences flanked by two TIR TEs belonging to the same element where the internal sequences were longer than 100 bp and shorter than 5,000 bp by following routine practices[67,88]. Because Pack-MULEs are short (~300 bp), 5,000 bp is sufficient to cover most *bona fide* Pack-TIRs. Taking humans as an example, the largest case is *ARF6* (3,585 bp). We also tried an alternative cutoff of 10,000 bp in humans and could not find additional Pack-TIRs.

To ensure that one TIR TE element recently captured a sequence with an unambiguous source, we implemented the following procedures using custom scripts based on Perl v5.26.2 Programming Language: (1) both flanking TIRs covered at least 10 bp of the terminals of the consensus TEs; (2) we searched the internal sequences against the genomes by running BLAT v35[89] with the parameters "minScore = 100, minIdentity = 90"; (3) to ensure that the capture event was recent and did not include additional deletions or insertions, we required the ratio of the length between internal sequences and parental copies to be between 80 and 120%; (4) we required that the top BLAT hit was unique such that the possible second hit had a lower score; and (5) we manually checked the candidates on the UCSC Genome Browser to exclude cases containing sequencing gaps in the parental loci. To ensure *bona fide* DNA TE-mediated duplication, we then excluded cases with parental copies harboring TEs in the flanking 100-bp regions to control for segmental duplications. We also discarded cases with parental copies encoding multiexonic genes and captured sequences harboring intronless copies to control for retroduplicates[90].

Herein, we followed the routine practice for Pack-MULEs in plants[30,67] and searched Pack-TIRs with parental copies not associated with TIR TEs to control for segmental duplications. Using such a pipeline, we may miss cases generated by the end bypass model for which TEs are linked with parental copies.

Our initial search found the multicopy family in Fig. 1d. We further searched all 370 candidate Pack-TIRs by lowering the cutoffs, e.g., incompleteness of TIR repeats. We then found another three multicopy cases whose members were slightly dissimilar to the one initially detected with our pipeline (see one example in Supplementary Fig. 18).

To date Pack-TIRs along the primate phylogenetic tree, we followed our previous practice[91,92] and deduced the orthology of Pack-TIRs based on the UCSC whole-genome syntenic alignments across primates. We inferred their evolutionary ages according to maximum parsimony.

We followed a previous method[10] to estimate whether microhomology could be observed by chance for 33 Pack-TIRs in humans. At the 66 breakpoints of these Pack-TIRs, 32 left- or right-side breakpoints harbored microhomology, whereas the others were blunt or linked with short de novo insertions and were therefore excluded in our simulation. For 32 cases, we took the parental copy together with its 5′ and 3′ flanking regions (half length of the parental copy) as the template (Supplementary Fig. 3a). We randomly selected 100 fragments from the parental locus with the same length as the captured sequence of the corresponding Pack-TIR and randomly selected the switching points in the consensus TIR TEs. We recorded the size of the microhomology at the breakpoint of interest and counted the proportion of replicates with equal or longer microhomology as the empirical *P*-value. The microhomology length distribution of 100 replicates is depicted as boxplots in Supplementary Fig. 3b. Taking case #02 as an example, only six replicates showed equal or longer microhomology than the length of observed microhomology, and we thus labeled this example as "$P = 0.06$".

Regarding the positions of breakpoints, because they can be closer to either the 5′ or 3′ end, we normalized them to the 5′ side. For example, if the relative position (proportion) of the breakpoints are 0.8–0.9, they would be normalized as "1-0.9"–"1-0.8" = 0.1-0.2. To test against the null (excision) model via transposition, we conservatively assumed the breakpoint closer to the terminal as the DSB site and thus recorded a value of 0.1. We generated 281 (the actual total number of Pack-TIRs) random samples of breakpoints based on uniform distributions.

### Identification and analyses of polymorphic Pack-TIRs in flies.
We searched for polymorphic Pack-TIRs (Supplementary Fig. 20b) based on our PacBio data across six DGRP lines (RAL-208, -379, -399, -427, -517, and -799). First, we mapped the long subreads to the reference genome (UCSC dm6) via BLASR v3.1.1.142244[93]. We detected structural variations via SMRT-SV v1[94] and then locally assembled them as contigs via Canu v1.8[95]. By searching against TEs annotated in Repbase[96], we analyzed whether insertions represented a candidate Pack-TIR, i.e., a fragment flanked by TIR TEs. We discovered only one case, *Ssk-FB4*. It should be noted that FlyBase[97] annotated two transcripts for *Ssk*, and the first intron was alternative. We used the major isoform according to expression sequence tag data from the UCSC Genome Browser[32]. By mapping the *Ssk-FB4*-containing contigs to the reference genome, we detected two insertion sites of *Ssk-FB4*, i.e., chrX: 2,745,191 and chr3R: 17,673,798. We also found that *Ssk-FB4* at chr3R: 17.7 Mb was further tandemly duplicated in the RAL-379 line. The duplicated copy contains a 6-bp deletion (Supplementary Fig. 14a), which was used in genotyping.

For the genotyping of individual *Ssk-FB4* copies across 206 DGRP lines[33,98], we downloaded Illumina resequencing data of DGRP and identified discordantly mapped read pairs and split reads, for which one read or one fragment was mapped to *Ssk* and the other one was mapped to *FB4*. We took such reads as evidence

supporting the presence of *Ssk-FB4* in a line of interest. Through this process, we found a third insertion site at chr3R: 14,340,248. We also noticed that *Ssk-FB4* was heterozygous at some sites, where we found reads continuously spanning the insertion site as well as discordant or split-mapped reads (Supplementary Data 5). Consistently, heterozygous sites are always situated in known inversions of DGRP lines[98]. Analogously, we performed genotyping in 90 GDL lines[39] and did not find any new insertion sites (Supplementary Data 6).

To infer the evolutionary history of *Ssk-FB4s*, we first curated full-length sequences of *Ssk-FB4*. Specifically, our lab maintained 20 DGRP lines with nine lines containing *Ssk-FB4* copies, and four of these lines were sequenced using PacBio technology. For the remaining five lines, we performed PCR for *Ssk-FB4s* using primers detailed in Supplementary Data 8 and then performed Sanger sequencing. In total, we generated 12 full-length sequences of *Ssk-FB4s* (GenBank accession numbers MT433937-MT433948) in these nine lines, which cover all four *Ssk-FB4* copies (Fig. 3c). We then aligned the internal sequences of 12 *Ssk-FB4s* and two orthologous *Ssk* in the reference genomes of *D. melanogaster* and *D. simulans* (UCSC droSim1) via MUSCLE v3.8.31[99]. We manually polished the multiple sequence alignment using MEGA v7.0.26[100]. We used RAxML v8.2.12[101] with the frequently used GTRGAMMA nucleotide substitution model[102] to construct a phylogenetic tree. To determine which of six full-length *FB4s* mediated the formation of *Ssk-FB4*, we performed similar phylogenetic analyses with the TIR TEs of *Ssk-FB4* and *FB4s*. In addition, the sequencing data shown in Supplementary Data 8 matched our genotyping results (100% accuracy).

Using the same PacBio data, we examined how six *FB4* loci harbored by the reference genome differed in the DGRP population. The sequences of all these loci were also submitted to GenBank with the accession numbers MT433949-MT433961.

**Searching signal of positive selection**. We deployed three strategies to detect positive selection acting on *Ssk-FB4s*. First, we calculated the $K_a/K_s$ ratio (the ratio between the nonsynonymous and synonymous substitution rates) for *Ssk* in 12 *Drosophila* (Supplementary Fig. 14e) using the codeml tool, which is part of the PAML v4.9h package[56]. Moreover, with *Ssk* as the outgroup, we used codeml to count the number of polymorphic substitutions along each lineage of *Ssk-FB4s* (Supplementary Fig. 14d). We further divided the protein encoded by *Ssk* or *Ssk-FB4* into three functional categories (transmembrane, intracellular and extracellular, Supplementary Fig. 14a) according to a previous study[38], and analyzed the enrichment of synonymous and nonsynonymous changes within each category.

Second, if the spread of *Ssk-FB4s* was under positive selection, their frequencies were expected to be higher than that of *FB4s*. We tested this hypothesis based on two sets of allele frequency data of TEs[103,104] segregated in DGRP. The two datasets were used in Fig. 5d and Supplementary Fig. 15a, respectively. Note that because the DGRP data were generated by sequencing hundreds of individuals and because individuals could differ between each other in terms of genotype, we simply counted the proportion of lines with *FB4s* or *Ssk-FB4s* as the frequency.

Third, if *Ssk-FB4s* rapidly spread under positive selection, selective sweep in which the linked genetic diversity would be lower could occur[58]. To test this hypothesis, we divided DGRP lines into one group harboring *Ssk-FB4* and another group not harboring *Ssk-FB4*. By implementing vcftools v0.1.12[105] with a 1,000-bp window size and a 500-bp step size, we took advantage of DGRP SNPs and calculated the nucleotide diversity (π) for 50-kb regions surrounding the insertion site in the two groups. We found that X-linked and 3R-linked (chr3R: 17.7 Mb) *Ssk-FB4s* were associated with a lower π on the left side (10 kb) of the breakpoint and on the two sides (15 kb on one side, 30 kb in total) of the breakpoints, respectively. For both loci, we recorded the ratio of π between the two groups as the observed value. To estimate whether neutral evolution resulted in such a deviation, we used scripts (R v3.4.4) to randomly sample 1,000 synonymous sites with the same chromosome, a similar (95%–105% fold) allele frequency and a similar recombination rate (95%–105% fold) as the two *Ssk-FB4* loci and examined whether a decrease in π across 10 or 30 kb could be observed. We counted the percentage of replicates as empirical *P*-values for which the ratio was equal to or lower than the observed values.

**Transcription analyses**. To determine the gene structure of Pack-TIRs in humans, we selected one public high-depth and strand-specific RNA-seq dataset[106] and used the splice-aware mapper STAR v2.4.0k[107] to map reads against the human GRCh38 genome under the guidance of the Ensembl v98 annotation. We selected STAR because it could differentiate different paralogs[108,109]. We retained only uniquely mapped reads with the criteria NH:i:1 in the BAM files[110]. We conservatively required at least five reads to define the expressed Pack-TIRs. We retrieved reads mapped to the Pack-TIRs and 100-kb flanking regions and performed de novo assembly using Trinity v2.6.5[111] with the parameter "--SS_lib_type RF" because the RNA-seq data are strand-specific. As a result, we should be able to infer the overall exon/intron structure of transcripts containing Pack-TIRs. We further aligned the assembled contigs to the genome via BLAT[89].

Our analyses in worms yielded results similar to those obtained in humans with the following exception: (1) worm data are strand-nonspecific[112] and the Trinity parameter was changed accordingly; and (2) to confirm the chimeric structure of *rpb-4*, we mapped the long-read (Nanopore) full-length transcriptome[113] using Minimap2 v2.15[114].

We used multiple datasets to quantify the expression levels of *Ssk*, *Ssk-FB4s*, *mesh*, and *Tsp2A*. First, for both the RNA-seq data of the guts from 38 DGRP lines and the data of the whole body from 200 DGRP lines[115,116], we directly used the processed data, i.e., FPKM (fragments per kilobase of exon model per million reads mapped). Because these two datasets did not differentiate *Ssk* and *Ssk-FB4s*, we interpreted the expression value of *Ssk* as the total expression of *Ssk* and *Ssk-FB4s*. Second, based on our own Illumina RNA-seq data across five tissues in six lines, we quantified the expression (TPM) in each line using Kallisto v0.44.0[108]. With the addition of sequences of *Ssk-FB4s* and *FB4s* in the transcriptome, Kallisto differentiated these paralogs via a mapping-free strategy. In parallel, we performed STAR-based alignment and loaded the mapping files into Integrative Genomics Viewer (IGV)[117] to visualize the gene structure of *Ssk-FB4s*.

**Analyses of DNA interaction data**. To detect the interaction between *Ssk* and *FB4s*, we attempted to analyze public Hi-C datasets in germline and embryogenesis for which mutations could be passed to offspring. We could only retrieve one published Hi-C dataset generated in embryogenesis[118]. We used Trim Galore v0.5.0 (http://www.bioinformatics.babraham.ac.uk/projects/trim_galore/) to perform quality and adapter trimming with the default parameters. We then mapped reads to the UCSC dm6 reference genome via HiCUP[119], removed experimental artifacts and PCR duplicates, and retained the high-quality mapping reads. We then extracted Hi-C paired contacts and implemented Juicer v1.8.9[120] to generate the interaction matrix with the normalization parameter "VC" and a 10-kb bin size as recommended in the literature[121,122]. We used custom scripts (Python v2.7.15) to calculate the interaction intensity.

**Protein identification, quantification, and co-immunoprecipitation (Co-IP)**. For proteomics, each sample including 10 midguts or 10 heads was lysed with the PCT-MicroPestle in 30 μL of lysis buffer (8 M urea and 0.1 M ammonium bicarbonate) supplemented with complete mini EDTA-free protease inhibitor (Roche) in a barocycler (2320EXT, Pressure BioSciences, Inc.)[123]. The lysis program contained 90 cycles, and each cycle comprised 25 s of high pressure at 45,000 psi and 10 s of ambient pressure at 30 °C. The protein concentration was determined using the Pierce BCA protein assay kit. After incubation with SDS-PAGE loading buffer in a boiling water bath for 10 min, the protein was separated by SDS-PAGE.

The other proteomic procedures followed routine practice. Specifically, for in-gel digestion, the gels were visualized by Coomassie blue staining. Protein bands with molecular weights in the range of 15–20 kDa (the mass of *Ssk* or *Ssk-FB4*, ~17 kDa) were cut for in-gel digestion with trypsin. The digested samples were lyophilized and further suspended in MS buffer before subsequent liquid chromatography tandem mass spectrometry (LC-MS/MS) analysis. The samples were spiked with 15% iRT peptides (Biognosis) and analyzed using a Q Exactive HF X Orbitrap mass spectrometer coupled with an EASY-nLC 1200 system. The peptides were loaded into the trap column for desalination and then washed into the analytical column (ReproSil-Pur Basic C18, 1.9 μm, 100 μm × 20 cm) for separation. Solvent A was 0.1% formic acid/2% acetonitrile/98% water, and solvent B was 0.1% formic acid/98% acetonitrile/2% water. A 30-min (60 min for the RAL-379 line) gradient from 3 to 35% solvent B at 450 nL/min was used for separation. The MS was operated in the targeted parallel reaction monitoring (PRM) mode, which was implemented in Skyline v19.1.0.193. MS scans were implemented for the mass range of 400–1500 *m/z* at a resolution of 60,000. Targeted ions were fragmented at a normalized collision energy of 28 for higher-energy collisional dissociation with an isolation window of 1.6 *m/z*. The MS/MS scans were acquired at a resolution of 30,000 with a fixed first m/z of 100 *m/z*. The maximum injection times for full MS and MS/MS scans were 50 and 200 ms, respectively. The automatic gain control target value was set to $1.0 \times 10^6$ and $5 \times 10^4$ for full MS and MS/MS scans, respectively.

PRM-MS raw files were processed in Skyline to generate an extracted ion chromatogram (XIC) and perform peak integration. The top five most intense fragment ions were used to quantify the peptide peak area. The sum of all identified peptide intensities was used as the protein intensity. To perform intensity-based absolute quantification (iBAQ), protein intensities were further divided by the number of theoretically observable peptides[53].

Co-IP assays were performed using the Pierce Crosslink Magnetic IP/Co-IP kit. The midguts were lysed in IP lysis buffer, and the debris was removed by centrifugation at $12,000 \times g$ and 4 °C for 20 min. The antibody against Mesh (diluted at 1:100) and A/G beads were added to the protein lysates. Afterward, the lysates were incubated overnight at 4 °C. The immune complex samples were washed five times with precold IP lysis buffer. All the samples were then boiled in SDS-PAGE loading buffer and analyzed by SDS-PAGE and in-gel digestion. Analogous PRM-MS runs were then performed as mentioned above.

**GWAS data analyses**. Since *Ssk-FB4s* could be involved in gut homeostasis or immune response, we performed a literature survey about public GWAS studies focusing on these traits in DGRP lines and found 13 studies (Supplementary Data 9). Considering the moderate population frequency of *Ssk-FB4s*, we focused on only four datasets with sufficient statistical power (more than 80 lines). These datasets examined the viability of different DGRP lines upon the infection of the West Nile virus subtype Kunjin (WNV-Kun)[71], the bacterium *Pseudomonas*

*entomophila*[72] or *Pseudomonas aeruginosa* Pa14[73] and the fungus *Metarhizium anisopliae* Ma549[73]. We divided the DGRP lines into two groups based on the presence or absence of *Ssk-FB4s* and compared the viability between these two groups.

## Data availability

All Sanger sequencing data generated in this study are available at the NCBI GenBank database under the accession codes MT433937-MT433961. The genome sequences and TE annotations for the 100 species are available at the UCSC Genome Browser database (https://genome.ucsc.edu).

## Code availability

The code produced for this study is available at https://github.com/clamp131/pack-TIR and archived in Zenodo [124].

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

## Acknowledgements

We thank Drs. Manyuan Long, Jinfeng Chen, Zongzhao Zhai, Qing Li, Yan Li, Tian Tang, Michael Stadler, and the Zhang laboratory members for the helpful discussions. We appreciate the Mesh antibody provided by Drs. Mikio Furuse and Yasushi Izumi. This research was supported by grants from the National Key R&D Program of China (2018YFC1406902, 2019YFA0802600), the Chinese Academy of Sciences (ZDBS-LY-SM005, No. XBZG-ZDSYS-201913), the National Natural Science Foundation of China (31701092, 31771410, 31970565, 91731302), the Youth Innovation Promotion Association of CAS (No. 2018112), and the third round of public welfare development and reform pilot projects of Beijing Municipal Medical Research Institutes (Beijing Medical Research Institute, 2019-1). Additionally, the C.L.C. lab is supported by grants from the I. Curie YPI program, the ATIP-Avenir program from CNRS and Plan Cancer, the Agence Nationale pour la Recherche (ANR) and the Institut National du Cancer (INCa). G.L.B. is supported in part by the Office of the Director, National Institutes of Health, under Award Number P51OD011106. The computing was jointly supported by the HPC Platform of BIG and that of the Scientific Information Centre of IOZ.

## Author contributions

Y.E.Z., S.T. and C.L.C. conceived and designed the study; S.T. and H.M. performed the computational analyses with help from D.W., Z.Z., and H.C.; J.W. and M.W. performed the experiments with help from M.X.W., H.Y., Y.Z., J.S, X.Z., S.G., J.M.W., and X.H.; and Y.E.Z., S.T., C.L.C., H.M., J.W., M.W., and G.L.B. wrote the manuscript.

## Competing interests

The authors declare no competing interests.
