## [Peer Review File · Nature Communications]

REVIEWER COMMENTS

Reviewer #1 (Remarks to the Author):

Comments to authors

Review for “DNA transposons mediate gene duplications in metazoans”

This study characterized the DNA transposons carrying gene or gene fragments in animals, which are called Pack-TIRs. Those include their copy number, distribution, selection, expression and function. The analysis led to three novel and important findings: 1) gene duplication by DNA transposons are prevalent in animals; 2) the duplication is caused by two distinct mechanisms, and the authors proposed a novel mechanism called “transposition after replication fork switching”. Certainly this model needs to be further tested in future, but at least they made the first step. 3) The function of one Pack-TIR element is experimentally tested in *Drosophila*, and it turned out this element is very important for the viability of flies. As a result, I have to say this study provides new insight into the mechanism and impact of gene duplication by DNA transposons, which is emerging as an important mechanism for generation of new genes. However, I do think the manuscript requires further clarification or discussion.

Major comments

1. The “gap filling” model

My main problem with the manuscript is that the authors have mixed two distinct mechanisms into a single gap filling model. One is the gap repair process following excision of an active element, this will only occur to an currently active element; the other is the repair process followed by DSB caused by other mechanisms. The second one will occur to any element or likely any sequence albeit the frequency may vary. I consider the two mechanisms are quite different because a) when an element is active, the frequency of excision is quite high. In contrast, the chance for other type of DSBs to occur within a small element should be relatively low; b) when excision occurs, the cut site is at the end or close to the end, so the conformation of the gap may not be the same as DSB created by other processes; c) most importantly, if an element is still be able to excise, it should be able to insert elsewhere and increase copy number too, which means the new born element with duplications could have multiple copies in the genome, which is different from what the authors have implied here; d) for the gap repair following excision, although the repairing itself does not require transposition, it requires excision to initialize the process, so it is transposition-dependent in some way. For all of the above reasons, I strongly suggest the authors distinguish these two mechanisms.

2. Single copy vs multiple copy element

As the analysis in this study indicated, the single copy and multiple copy elements have different origin or formed through different mechanisms. Certainly, if an element is formed through transposition-independent mechanism and the relevant family lost activity, it will remain for one copy. However, if we reverse the logic, stating all single copy elements are formed by transposition-independent mechanism, that is unlikely right. For example, in maize genome, a few family of LTR retrotransposons achieved very high copy number and contributed to expansion of genome size. In contrast, the majority of the families have only one or two copies (please see R.S. Baucom, J.C. Estill, C. Chaparro, N. Upshaw, A. Jogi, J.M. Deragon, R.P. Westerman, P.J. Sanmiguel, J.L. Bennetzen, Exceptional diversity, non-random distribution, and rapid evolution of retroelements in the B73 maize genome, *PLoS Genet.* 5 (2009) e1000732). I don't think you could conclude that all those families are formed by transposition-

independent mechanisms – they have low copy numbers for other reasons, either their activity is low or they are selected against. My point is, it is not impossible that all or most single copy Pack-TIR elements are from transposition-independent mechanism, but this should not be generalized. I think this should be clarified/discussed in the manuscript.

Minor comments

1. Figure 1D. In the left panel, blue color represents multiple copy and black represents single copy. However, in the right panel, black represent transposon sequence (high copy) and blue represent the gene sequences (low copy). I got very confused at the beginning. My suggestion is to change the color of one panel so to reduce the level of confusion.

2. Figure 2E. I think the detailed position should be given for the gene and TE in Gibbon. The current figure looks like they are in the exactly same place, which I guess it is not the case.

3. Row 233 “a solo TIR TE”

This is confusing too. To my understanding, “a solo TIR TE” means a truncated TE with a single terminal sequence, the other end is gone. Obviously that is not the case here. So please try to revise. Maybe you could call it “a homologous TIR TE without duplication”?

4. Row 336 – 337 “First, the size of duplicated regions is small with a median of 224 bp (Fig. 4A),”

I am just curious whether there is a difference between single copy and multiple copy elements?

5. Row 400 – 404 “The former case is worth noting where a part of 3’ UTR and the terminal coding exon of F43E2.6 was copied to the upstream of the nearby essential gene, i.e., RNA polymerase II subunit (rpb-4). Short- and long-read RNA-seq data reveal two isoforms with Pack-TIRs providing alternative 5’ UTRs (Fig. 4E). Thus, similar to GTF2H5 and H2BC15, the regulation of rpb-4 is likely changed.”

Based on Figure 4E, there is a gene between the TE and the rpb-4 gene, so I am not sure how the authors know “the regulation of rpb-4 is likely changed”? If you don’t have other evidence to support the statement, please turn the tone down.

6. Figure 5G, the comparison is between with and without the Ssk-FB4. I guess the pink refers to “without” and the blue refers to “with”. Nevertheless it would be good to tell readers which one is which.

Reviewer #2 (Remarks to the Author):

Summary

While retrotransposons have long been implicated in amplification and transduction of host genes in many organisms, the ability of DNA transposons to transduce gene fragments in animals is comparatively understudied, as opposed to plants where Pack-MULEs and Helitrons have been extensively examined. In this paper, the authors investigate the frequency of DNA transposon mediated capture and transduction/duplication of host genic sequences and whether or not these captured fragments may be under selection for novel host function. To do this, they survey 100 animal genomes

as well as population data from *Drosophila melanogaster* to identify gene fragments flanked by terminal inverted repeat sequences, dubbed "Pack-TIRs." They identify several candidate Pack-TIRs, most of which are single copy, and use evolutionary analysis to assess the origin of several Pack-TIRs in primates, frog, and *D. melanogaster*, and based on these analyses they propose that Pack-TIRs form primarily via a "gap-filling" or a model where replication fork stalling leads to template switching and subsequent transposition (FoSTeST). They then investigate whether these DNA TE transduced gene fragments may be functional using a combination of transcriptomic, proteomic, and selection analyses. They especially focus on the multicopy Ssk-FB4 Pack-TIR family in *D. melanogaster*, and show they are expressed both at the RNA and protein level, and that knocking out one copy leads to reduced survival rate of KO flies relative to WT controls. Based on this data, the authors conclude that DNA transposons, particularly of the TIR type, represent a previously unappreciated source of genetic variation by capturing host gene fragments, and that they can do so in transposition-independent mechanisms.

General Comments

The authors address an interesting biological question and expand upon previous documentation that TEs are capable of transducing host gene fragments. Although the concept is not entirely new, the authors investigate TEs that have not been typically implicated in this process and identify possible mechanisms of gene capture that are independent of transposition, which is particularly interesting. Most of the analyses seem solid, and the figures are aesthetically beautiful and easy to understand.

However, the paper as written is difficult to read for a number of reasons: 1) the language is not sufficiently precise, 2) the authors use several unusual phrasings that make it difficult to understand the meaning, and 3) there is not adequate experimental detail in the main body of the text to understand the analyses being performed. I have listed some examples in the specific comments below, but I strongly recommend a round of editing to improve clarity. The authors should also be careful not to conflate transcription/splicing/translation of Pack-TIRs with functionality. Such conclusions are better supported by the evolutionary and selection analyses.

There are also potential issues with some of the analyses. For example, when determining if a PackTIR is conserved within or between species, both in the text and in tables, it is unclear whether absent means that the two TIRs are present but they have not captured the sequence, or whether the TIR transposon is absent altogether (including evidence of either an empty site or excision footprint). This information is necessary to accurately interpret the evolutionary history of each Pack-TIR.

The main text section on selection analyses is also very confusing as written. I think the authors intend to argue that Ssk-FB4 copies are under positive selection because nonsynonymous mutations are enriched in Ssk-FB4 functional domains, but this is not immediately apparent. Additionally, it is very unclear what the authors mean by "negative selection where 65.8% of elements are singletons," as such an observation could also be attributed to a low transposition rate. It is also unclear to me why a higher frequency of Ssk-FB4 alleles in the population relative to other FB4 copies is necessarily indicative of positive selection. The method section is easier to understand, but the results are hard to evaluate and should be rewritten to make the rationale, logic, and methods used clearer.

Finally, regarding the KO experiment, it appears that the authors only deleted the 4 base pair insertion

unique to the X chromosome Ssk-FB4 copy, rather than the whole locus. It is unclear to me why the authors chose this strategy, and whether or not this 4 base pair deletion would be sufficient enough to abolish X-Ssk-FB4 function, if any. Similarly, the KO experiments are hard to interpret, as it is not clear whether the observed effects are due to the 4bp deletion or to CRISPR off targets. Thus, the authors should 1) elaborate on the rationale behind their KO design and 2) either perform a rescue experiment or temper their conclusions to say that the results of the viability assay are consistent with an effect due to loss of function of the X-Ssk-FB4 copy.

Overall, I think that, once these issues with the writing and experimental detail are resolved, the paper will be a useful contribution to the field.

Specific Comments

- Line 51: “renovations” is an unusual word choice; typically “innovations” is used
- Line 54-55: Here and elsewhere in the paper the authors argue that the mechanism of gene duplication affects their functional trajectories, which in principle is probably correct, but I don’t think statements like “duplicates generated by TEs are more likely neofunctionalized due to various mechanisms” is accurate
- Line 71: Here the authors should be clear what they mean by “spatial proximity” – ie: linear or 3D proximity?
- Lines 157-159: While it is certainly possible that a linear correlation between the # of PackTIRs and the total number of TIRs in the genome could suggest a transposition independent mechanism, it could equally just be due to the fact that the presence of more TE copies increases the overall opportunity to generate a PackTIR
- Line 177: “...21 cases merged from orthologs” It is unclear what the authors mean by “merged from orthologs” both here and in the methods
- Line 182: Here and throughout the paper the authors use “dominance” when I think it would be better to use “common” or “most common” as dominance has a very particular genetics definition
- Line 236: What do the authors mean by “polarize the evolutionary order”?
- Line 252-254: Do the authors observe an empty site in the African clawed frog or is it just absent from the assembly?
- Line 320-322: The conclusion presented here is too strong for the data – the scenario the authors propose is only one possibility
- Lines 335-347: For the transcriptome analysis, the authors should elaborate from which promoter the gene fragments are transcribed from. The TE? An upstream/downstream gene? A cryptic promoter?
- Lines 427-428: The authors should be clear “30% or more” PackTIRs are transcribed in these three species, and that further studies are necessary to determine if this is generalizable across all animal PackTIRs
- Line 466: “Since translation is not equal to function affecting fitness” is grammatically awkward
- Line 467: The authors should be precise about what they mean by “evolve under natural selection”
- Lines 492-495: It is not clear to me why an absence of interaction between X-Ssk-FB4 protein and mesh/Tsp2A would be expected to affect the expression of these two genes. The authors should elaborate more on the mesh/Tsp2A angle so that the reader can better understand how their data fits into known roles for these genes
- Line 566: It is not clear what the authors mean by “conferring side function”

- Lines 588-591: The authors limit their pipeline to only identify short genic insertions – did the authors ever find larger insertions?
- Lines 622-629: Overall the method to simulate the homology breakpoints is not clear – the authors should elaborate on this point
- Line 680: Is 2 generation outcrosses enough to control for background mutations? Seems like more backcrosses should have been done
- Lines 728-732: The authors should elaborate a bit more on how they performed de-novo transcriptome assembly, including which reads were included in the model – the text makes it seem that only the PackTIR reads were included which, if the case, could lead the authors to miss the overall exon/intron structure of transcripts containing PackTIRs

Reviewer #3 (Remarks to the Author):

Tan and colleagues present comprehensive analysis of DNA-transposon-mediated gene duplications in metazoan. This is very interesting analysis and worth consideration by Nature Communication. However, there are several issues that should be resolved before the manuscript can be published.

Firstly, the title is a bit misleading because most of presented evidence are pointing to pieces of genomic DNA, not the real genes that were duplicated with aid of transposon action. I suggest that the title of the manuscript should be modified accordingly to better reflect the main findings.

Another issue is functional analysis of duplicated genes and Ssk-FB4s in particular. Although, knock out experiments and evolutionary analysis showed that Ssk-FB4s gene is essential and its function is different from the source gene (Ssk), the authors didn't demonstrate what the new ruction could be. I would very much appreciate the discussion, even speculative one, on the potential function of Ssk-FB4s. Additionally, a comprehensive comparison of proteins coded by Ssk-FB4 and Ssk genes would enhance the discussion of this interesting gene.

Finally, I would love to see discussion/comparison of the described here phenomenon to DNA transduction mediated by L1 and SVA elements described previously by several groups.

Minor pints and clarifications.

1. The authors should pay better attention to detail description of the figures and avoid discrepancies in the presented results. For instance, in Figure 1 A we are presented with information on distribution of Pack-TIRs in 55 animals with some of the species having no Pack-TIRs detected. However, in the beginning of the paragraph we can read that 100 animal genomes were analyzed and Pack-TIRs were found within 55 genomes (line 109 and 120, respectively). Additionally, in the same figure legend we can read "To simplify, 18 primates and 12 Drosophila species were represented by human and D. melanogaster, respectively." I understand the authors intention but why not to label the respected phylogenetic tree leaves with "primates" and "Drosophila?" Interestingly, there is no D. melanogaster

label but fruitfully that most likely represents *Drosophila* lineage.

What colored numbers in Figure 3C represent?

2. "Microhomology" term. Homology is an evolutionary term that indicates common ancestry of two features, e.g. piece of DNA. However, the authors use this term to refer to two or more short similar sequences without implying their common evolutionary ancestry. I strongly suggest to avoid the microhomology term throughout the manuscript unless the authors would like to indicate their common ancestry, which I believe is not the case.

3. Lines 448-450. "The lower nonsynonymous data is based on the whole *Drosophila* genus to increase the statistical power (Fig. S12E), while the synonymous data is only based on the *melanogaster* subgroup to avoid the saturation of synonymous changes." I'm not sure if this approach is methodologically acceptable. Altering data sets to get better statistical support is a dangerous operation as one can easily lose the objectivity of data analysis.

INDEX OF THE RESPONSE LETTER

The major changes of the manuscript	3
Comments of Reviewer 1.....	5
General comments	5
Major comments	5
1. The “gap filling” model	5
2. Single copy vs multiple copy element	9
Minor comments	10
1. Figure 1D	10
2. Figure 2E.....	11
3. Row 233 “a solo TIR TE”	11
4. Row 336 – 337	12
5. Row 400 – 404.....	13
6. Figure 5G	13
Comments of Reviewer 2.....	14
General comments	14
Major comments	14
1. However, the paper as written is difficult to read for a number of reasons.....	14
2. There are also potential issues with some of the analyses	15
3. The main text section on selection analyses is also very confusing as written	18
4. Finally, regarding the KO experiment	21
Minor comments	27
1. Line 51	27
2. Line 54-55.....	27
3. Line 71	27
4. Lines 157-159	27
5. Line 177	28
6. Line 182	28
7. Line 236	28
8. Line 252-254.....	31
9. Line 320-322.....	31
10. Lines 335-347	31
11. Lines 427-428	32
12. Line 466	32
13. Line 467	32
14. Lines 492-495	32
15. Line 566	33
16. Lines 588-591	33
17. Lines 622-629	34
18. Line 680	34
19. Lines 728-732	35

Comments of Reviewer 3.....	36
General comments	36
Major comments	36
1. Firstly, the title is a bit misleading.....	36
2. Another issue is functional analysis of duplicated genes and Ssk-FB4s in particular	36
3. Finally, I would love to see discussion/comparison of the described here phenomenon to DNA transduction mediated by L1 and SVA elements described previously by several groups	38
Minor comments	38
1. The authors should pay better attention to detail description of the figures and avoid discrepancies in the presented results.....	38
2. What colored numbers in Figure3C represent?	39
3. “Microhomology” term.....	40
4. Lines 448-450	40
References.....	41

The major changes of the manuscript

We deeply appreciate the referees' constructive comments. Over more than six months, we have made extensive changes to our manuscript in response to the referees' comments. We have addressed each comment below, where table and figure IDs refer to the new version. To facilitate an overall understanding, we summarize major points here.

As the referees pointed out, our manuscript mainly reported novel TIR TE-mediated mechanisms of duplication, which are present across different animal lineages. The manuscript additionally used proteomic, evolutionary and functional approaches to analyze the function of TIR TE-mediated duplicate genes especially *Ssk-FB4s*. The new version of our manuscript has been significantly improved in these major and minor aspects, as has the general presentation.

1. Motivated by the comments from referee 1 on mutational mechanisms with specific predictions, we performed additional analyses (*e.g.*, the distribution and size of gaps in TEs; Figs. 2F-G). Our previous and new data consistently support the prevalence of a gap-filling process induced by broken fragile sites associated DSBs, which is transposition-independent. Furthermore, in consideration of a comment by referee 3, we present our discovery in a broader context of TE-mediated duplication by discussing the common features of duplication mechanisms mediated by various types of transposons. Given these changes, we revised the title to, "DNA transposons mediate duplications via transposition-independent and -dependent mechanisms in metazoans".
2. Referee 2 was concerned with potential off-targets and suggested that we either perform rescue experiments or tone down conclusions. To produce the best possible manuscript, we tried our best to generate a rescue line but unfortunately failed. We then performed whole genome sequencing to identify off-target mutations, and repeated phenotyping in lines with additional outcrosses. Although sequencing data analyses indicated the absence of off-targets, the phenotypic effect was not constant. Since we cannot dissect *Ssk-FB4s*' functionality in the near future, and as this manuscript is principally about the mutational mechanism, we removed the section of viability analysis and tone down our statements. Nonetheless, phenotypic inconstancy suggests that *Ssk-FB4s*' function is related to environmental conditions. Consistently, we found that *Ssk-FB4s* are associated with variable but always stronger resistance against distinct microbial infections (Fig. S19).
3. Motivated by the comments of all referees, we modified the presentations with precision (*e.g.*, Figs. 1D and 5C), and we added multiple illustrative figures for better understanding, *e.g.*, how we evaluated the length of microsimilarity (microhomology, Fig. S3A), or how we performed selection analyses (Fig. S13). Additionally, we briefly reiterated how we performed the analyses in the Results section, to enhance readers' understanding.

Taken together, we believe that this new version is strongly enhanced in terms of mechanistic depth, methodological stringency and readability.

To most clearly present new information, we revised the text, added new supplemental materials, and improved the writing. For the convenience of the editor and referees, we are uploading a version free of tracked changes and another with the major changes marked in light blue.

Reviewer #1 (Remarks to the Author):

Comments to authors

Review for “DNA transposons mediate gene duplications in metazoans”

*This study characterized the DNA transposons carrying gene or gene fragments in animals, which are called Pack-TIRs. Those include their copy number, distribution, selection, expression and function. The analysis led to three novel and important findings: 1) gene duplication by DNA transposons are prevalent in animals; 2) the duplication is caused by two distinct mechanisms, and the authors proposed a novel mechanism called “transposition after replication fork switching”. Certainly this model needs to be further tested in future, but at least they made the first step. 3) The function of one Pack-TIR element is experimentally tested in *Drosophila*, and it turned out this element is very important for the viability of flies. As a result, I have to say this study provides new insight into the mechanism and impact of gene duplication by DNA transposons, which is emerging as an important mechanism for generation of new genes. However, I do think the manuscript requires further clarification or discussion.*

Many thanks for your encouraging summary.

Major comments

1. The “gap filling” model

My main problem with the manuscript is that the authors have mixed two distinct mechanisms into a single gap filling model. One is the gap repair process following excision of an active element, this will only occur to an currently active element; the other is the repair process followed by DSB caused by other mechanisms. The second one will occur to any element or likely any sequence albeit the frequency may vary. I consider the two mechanisms are quite different because a) when an element is active, the frequency of excision is quite high. In contrast, the chance for other type of DSBs to occur within a small element should be relatively low; b) when excision occurs, the cut site is at the end or close to the end, so the conformation of the gap may not be the same as DSB created by other processes; c) most importantly, if an element is still be able to excise, it should be able to insert elsewhere and increase copy number too, which means the new born element with duplications could have multiple copies in the genome, which is different from what the authors have implied here; d) for the gap repair following excision, although the repairing itself does not require transposition, it requires excision to initialize the process, so it is transposition-dependent in some way. For all of the above reasons, I strongly suggest the authors distinguish these two mechanisms.

We deeply appreciate this series of insightful comments. We previously mixed these two mechanisms for the following reasons: 1) for both mechanisms, TEs capture a non-TE sequence via transposition-independent template switching; and 2) we could not determine how best to differentiate these mechanisms, because an active TIR TE could

also harbor DSBs induced by either excision or breakage of internal fragile sites. Now, guided by the referee's four constructive comments with specific predictions, we found that most Pack-TIRs were more likely generated by the gap-filling process induced by broken fragile sites rather than by excision. We thus revised the manuscript accordingly.

We modified the Introduction along with Fig. S1 to explicitly describe these two mechanisms:

“Second, *P* elements capture sequences under the gap-filling model²¹. In this model, double-strand breaks (DSBs) occur in two scenarios: 1) the internal sites are broken, possibly induced by secondary structures^{22,23} (Fig. S1B); or 2) complete TEs are excised due to transposition²⁴ (Fig. S1C). During the repair, the template could switch from the sister strand to adjacent external sequences (called fillers) in 3D proximity, leading to the capture of fillers^{21,25}. The whole process in the former scenario (Fig. S1B) is transposition-independent.” (Line 71-78)

“Figure S1. Known models on the mechanism through which TIR TEs capture sequences. A) End bypass model. B-C) Gap-filling model. Double-strand breaks (DSBs) occur within TEs due to fragile sites (B) or excision of active TEs (C), and this occurrence was followed by 5’-end resection by exonucleases and gap repair. The strand under repair might switch to a non-TE sequence. Internal TE sequences and fillers are marked in red and light blue, respectively. TIRs are shown by black arrows.”

We then tested all four predictions by the referee. Specifically, for comment a), we inferred that Pack-TIR associated TEs were generally inactive. Now, the Results and Table S1 read as follows:

“In actuality, the consensus sequences of TEs tend to be short, with a median length of only 374 base pairs (bp). Moreover, individual TE copies are different from the consensus by a median divergence value of 18.3% (Table S1). These two patterns suggest that TEs likely represent degenerate ancient relics.” (Line 165-169).

TE	TE superfamily	Consensus TE length (bp)	Median divergence
XR_XL	Kolobok	603	10.20%
POR	hAT	308	10.20%
VI_XL	PIF-Harbinger	487	14.80%
VI_XL	PIF-Harbinger	487	14.80%
JH12_XL	TcMariner	326	15.90%
POR	hAT	308	10.20%
POR	hAT	308	10.20%

A partial snapshot of Table S1.

For comment b), we examined the breakpoint positions of Pack-TIRs relative to the consensus TEs in Figs. 2F-G and updated the Results.

“Finally, we analyzed whether DSBs induced by breaks within TEs (Fig. S1B) or by TE excisions via transposition (Fig. S1C) initiated the gap-filling process. We expected that the abortive gap repair in the latter scenario would generate larger deletions with breakpoints biased to the terminals of TEs compared with those obtained in the former scenario. We found evidence compatible with the former scenario: the distances between two breakpoints only account for a median of 4.6% TE sizes (Fig. 2F), and the breakpoints were moderately but significantly skewed toward the internal region of TEs compared with the random uniform distribution (Fig. 2G, Methods). The skew is likely due to the enrichment of internal fragile sites.” (Line 267-275)

“Figure 2F. Proportion of each gap relative to the corresponding consensus TE. The distribution is shown as a violin plot where the black bar indicates the interquartile range, the violin curve indicates the probability density of the data, and the line in the middle indicates the median. **G) Cumulative density distribution of the observed and simulated relative positions of breakpoints to consensus TEs.** Values closer to 0.5 at X-axis indicate positions closer to the middle of the TEs, whereas values close to 0 indicate positions closer to the terminals.”

Regarding comment c), our prior data show the prevalence of single-copy Pack-TIRs which are closely linked with the parental copies (Figs. 2A-B), again supporting a transposition-independent mechanism (Fig. S1B).

After addressing comments a) to c), the transposition-independent gap-filling process is more strongly supported than in the previous manuscript. Moreover, Referee 3 mentioned that Pack-TIRs also captured intergenic sequences and thus the previous title was not precise (Page 36). We therefore changed the title to “DNA transposons mediate duplications via transposition-independent and -dependent mechanisms in metazoans” in order to better summarize the key findings of the new version.

Finally, in regard to comment d), we fully agree that, although the repairing itself does not require transposition, the mechanism in Fig. S1C requires excision via transposition and thus is transposition-dependent. We have thus made our writing more precise as follows:

“During the repair, the template could switch from the sister strand to adjacent external sequences (called fillers) in 3D proximity, leading to the capture of fillers^{21,25}. The whole process in the former scenario (Fig. S1B) is transposition-independent.” (Line 75-78)

“Third, the gap-filling process induced by the fragile site (Fig. S1B) might indicate a longer waiting time between TE insertion and non-TE sequence capture compared with that obtained with the process induced by active transposon excision (Fig. S1C). Thus, TIR TEs without duplications should be present at the orthologous locus of outgroup species if duplications are specific to the focal species or lineages.” (Line 252-256)

“Specifically, our results, particularly the findings that the emergence of Pack-TIRs when DNA TEs were inactive in primates (Fig. 1E and Table S3) and that the distribution of breakpoints was biased to internal regions (Fig. 2G), indicate that the transposition-independent gap-filling model (Fig. S1B) underlies the formation of most single-copy Pack-TIRs. Certainly, the transposition-dependent gap-filling model (Fig. S1C) might also generate a small proportion of single-copy Pack-TIRs.” (Line 546-551)

2. Single copy vs multiple copy element

As the analysis in this study indicated, the single copy and multiple copy elements have different origin or formed through different mechanisms. Certainly, if an element is formed through transposition-independent mechanism and the relevant family lost activity, it will remain for one copy. However, if we reverse the logic, stating all single copy elements are formed by transposition-independent mechanism, that is unlikely right. For example, in maize genome, a few family of LTR retrotransposons achieved very high copy number and contributed to expansion of genome size. In contrast, the majority of the families have only one or two copies (please see R.S. Baucom, J.C. Estill, C. Chaparro, N. Upshaw, A. Jogi, J.M. Deragon, R.P. Westerman, P.J. Sanmiguel, J.L. Bennetzen, Exceptional diversity, non-random distribution, and rapid evolution of retroelements in the B73 maize genome, PLoS Genet. 5 (2009) e1000732). I don't think you could conclude that all those families are formed by transposition-independent mechanisms – they have low copy numbers for other reasons, either their activity is low or they are selected against. My point is, it is not impossible that all or most single copy Pack-TIR elements are from transposition-independent mechanism, but this should not be generalized. I think this should be clarified/discussed in the manuscript.

We appreciate this insightful comment, particularly as it relates to the previous comment. The new analyses (Figs. 2F-G) further support that most single-copy Pack-TIRs were likely generated by the transposition-independent gap-filling mechanism (Fig. S1B). However, we do agree that some single-copy Pack-TIRs could be still generated by the transposition-dependent gap-filling mechanism (Fig. S1C) or FoSTeST model. Thus, we followed the referee's suggestion, cited the paper mentioned, and toned down the previous statement.

“Specifically, our results, particularly the findings that the emergence of Pack-TIRs when DNA TEs were inactive in primates (Fig. 1E and Table S3) and that the distribution of breakpoints was biased to internal regions (Fig. 2G), indicate that the transposition-independent gap-filling model (Fig. S1B) underlies the formation of most single-copy Pack-TIRs. Certainly, the transposition-dependent gap-filling model (Fig. S1C) might also generate a small proportion of single-copy Pack-TIRs. Their copy numbers did not increase, possibly due to low activity of the corresponding TEs or negative selection⁶².” (Line 546-553)

Minor comments

1. Figure 1D. In the left panel, blue color represents multiple copy and black represents single copy. However, in the right panel, black represent transposon sequence (high copy) and blue represent the gene sequences (low copy). I got very confused at the beginning. My suggestion is to change the color of one panel so to reduce the level of confusion.

Thanks and we accordingly changed the color as suggested by the referee.

“Figure 1D. Pack-TIRs in western clawed frog.”

2. Figure 2E. I think the detailed position should be given for the gene and TE in Gibbon. The current figure looks like they are in the exactly same place, which I guess it is not the case.

Thanks for this reminder. The parental copy and Pack-TIR are separated by ~1,300 bp. We followed the comment and now showed the specific positions.

“Figure 2E. A comparative view of a demo Pack-TIR in gibbon (case #144 in Table S1).”

3. Row 233 “a solo TIR TE”

This is confusing too. To my understanding, “a solo TIR TE” means a truncated TE with a single terminal sequence, the other end is gone. Obviously that is not the case here. So please try to revise. Maybe you could call it “a homologous TIR TE without duplication”?

Thanks for this wonderful comment. We actually thought about this issue on many occasions and could not think up a succinct but suitable term. We agree that “solo” is

often used to describe one terminal of TEs (e.g., LTR TEs). We have now followed your comment and revised all three sections with “solo”.

“Third, the gap-filling process induced by the fragile site (Fig. S1B) might indicate a longer waiting time between TE insertion and non-TE sequence capture compared with that obtained with the process induced by active transposon excision (Fig. S1C). Thus, TIR TEs without duplications should be present at the orthologous locus of outgroup species if duplications are specific to the focal species or lineages. We tested this hypothesis in primates due to their densely sampled phylogeny (Fig. 1E). The majority (91 out of 97, or 94%) fit the expected scenario where orthologous TIR TEs without duplications existed in the outgroup species (Table S3).” (Line 252-259)

“Taking Pack-*hAT-N8_XT* (Fig. 1D) as an example, all three derived Pack-TIRs or *hAT-N8_XT* without duplication were absent in the orthologous loci of the outgroup, the African clawed frogs (Fig. S5), which diverged from the western clawed frogs 57 Mya³⁷.” (Line 285-288)

“Taken together, the absences of orthologous TEs without duplication across multicopy Pack-TIRs in frogs, alligators and flies suggest that multicopy Pack-TIRs were likely generated through the FoSTeST process.” (Line 364-366)

4. Row 336 – 337 “First, the size of duplicated regions is small with a median of 224 bp (Fig. 4A),”

I am just curious whether there is a difference between single copy and multiple copy elements?

We have identified only four multicopy Pack-TIRs, and their lengths are 206, 279, 291 and 458 bp, respectively. We marked these four cases as red dots in Fig. 4A. Now the manuscript reads as follows:

“The size of multicopy Pack-TIRs appeared to be larger, although this difference was not statistically significant (Fig. 4A).” (Line 372-373)

“Figure 4A. Length distribution of the internal sequences. The distribution is shown as a violin plot in Fig. 2F. The four red dots represent the multicopy Pack-TIRs.”

5. Row 400 – 404 “The former case is worth noting where a part of 3’ UTR and the terminal coding exon of F43E2.6 was copied to the upstream of the nearby essential gene, i.e., RNA polymerase II subunit (*rpb-4*). Short- and long-read RNA-seq data reveal two isoforms with Pack-TIRs providing alternative 5’ UTRs (Fig. 4E). Thus, similar to *GTF2H5* and *H2BC15*, the regulation of *rpb-4* is likely changed.”

Based on Figure 4E, there is a gene between the TE and the *rpb-4* gene, so I am not sure how the authors know “the regulation of *rpb-4* is likely changed”? If you don’t have other evidence to support the statement, please turn the tone down.

We agree and have rephrased as follows:

“Thus, similar to *GTF2H5* and *H2BC15*, the gene structure of *rpb-4* was changed, which might have a functional consequence.” (Line 438-440)

6. Figure 5G, the comparison is between with and without the *Ssk-FB4*. I guess the pink refers to “without” and the blue refers to “with”. Nevertheless it would be good to tell readers which one is which.

We are sorry for this confusion. We took your advice and modified Fig. 5E.

“Figure 5E. Transcription of *Ssk*, *mesh* and *Tsp2A* in the gut across lines with or without *Ssk-FB4s*. The distribution is shown as violin plots. Because the expression profiles of *Ssk* and *Ssk-FB4s* are largely similar, we simply merged the expression of *Ssk* and *Ssk-FB4s*.”

Reviewer #2 (Remarks to the Author):

Summary

*While retrotransposons have long been implicated in amplification and transduction of host genes in many organisms, the ability of DNA transposons to transduce gene fragments in animals is comparatively understudied, as opposed to plants where Pack-MULEs and Helitrons have been extensively examined. In this paper, the authors investigate the frequency of DNA transposon mediated capture and transduction/duplication of host genic sequences and whether or not these captured fragments may be under selection for novel host function. To do this, they survey 100 animal genomes as well as population data from *Drosophila melanogaster* to identify gene fragments flanked by terminal inverted repeat sequences, dubbed “Pack-TIRs.” They identify several candidate Pack-TIRs, most of which are single copy, and use evolutionary analysis to assess the origin of several Pack-TIRs in primates, frog, and *D. melanogaster*, and based on these analyses they propose that Pack-TIRs form primarily via a “gap-filling” or a model where replication fork stalling leads to template switching and subsequent transposition (FoSTeST). They then investigate whether these DNA TE transduced gene fragments may be functional using a combination of transcriptomic, proteomic, and selection analyses. They especially focus on the multicopy Ssk-FB4 Pack-TIR family in *D. melanogaster*, and show they are expressed both at the RNA and protein level, and that knocking out one copy leads to reduced survival rate of KO flies relative to WT controls. Based on this data, the authors conclude that DNA transposons, particularly of the TIR type, represent a previously unappreciated source of genetic variation by capturing host gene fragments, and that they can do so in transposition-independent mechanisms.*

Many thanks for your comprehensive and precise summary.

General Comments

The authors address an interesting biological question and expand upon previous documentation that TEs are capable of transducing host gene fragments. Although the concept is not entirely new, the authors investigate TEs that have not been typically implicated in this process and identify possible mechanisms of gene capture that are independent of transposition, which is particularly interesting. Most of the analyses seem solid, and the figures are aesthetically beautiful and easy to understand.

We appreciate your positive and encouraging comments.

However, the paper as written is difficult to read for a number of reasons: 1) the language is not sufficiently precise, 2) the authors use several unusual phrasings that make it difficult to understand the meaning, and 3) there is not adequate experimental

detail in the main body of the text to understand the analyses being performed. I have listed some examples in the specific comments below, but I strongly recommend a round of editing to improve clarity. The authors should also be careful not to conflate transcription/splicing/translation of Pack-TIRs with functionality. Such conclusions are better supported by the evolutionary and selection analyses.

We regret that our logical flow was insufficiently smooth and precise. We improved the writing in four aspects by following your comments.

First, we accepted your Specific Comments and improved the presentation regarding precise writing and unusual phrasing (see below).

Second, for experimental details, we added several sentences in the main text to summarize how experiments were performed.

“We finally examined whether *Ssk-FB4s* can encode proteins. We collected protein bands with molecular weights in the range of 15 to 20 kDa (the mass of *Ssk* or *Ssk-FB4* is ~17 kDa) and used a mass spectrometer to search for peptides (Methods). We identified high-quality unique peptides encoded by three *Ssk-FB4* copies in available fly lines (Figs. 5A and S12, and Table S9).” (Line 470-474)

“Moreover, after co-immunoprecipitation with an antibody against Mesh, we quantified the protein intensity of *Ssk-FB4* and *Ssk*. Despite the 500% abundance of X-linked *Ssk-FB4* relative to that of *Ssk* in the fly midgut, the former abundance dropped to 70% of the latter in the immunoprecipitates (Fig. 5F).” (Line 533-536)

Third, regarding the functionality of Pack-TIRs, we fully agree that various biochemical activity is not equal to function subject to natural selection. We have now toned down the relevant sections and made the writing more accurate.

“We then inferred whether Pack-TIRs are potentially functional by analyzing their sequence and expression features.” (Line 369-370)

“*Ssk-FB4s* likely represent a functional protein family evolving under positive selection.” (Line 468-469)

“We finally examined whether *Ssk-FB4s* can encode proteins.” (Line 470)

“Thus, *Ssk-FB4s* appear to not be involved in the protein complex of *Ssk*/*Mesh*/*Tsp2A* or only weakly interact with this complex. Together with the contrastive substitution pattern (Fig. 5C), *Ssk-FB4s* and *Ssk* very likely have different functions.” (Line 537-539)

Finally, our coauthor, Dr. Graham L. Banes, who is a native English speaker, carefully polished the whole manuscript.

There are also potential issues with some of the analyses. For example, when determining if a Pack-TIR is conserved within or between species, both in the text and in tables, it is unclear whether absent means that the two TIRs are present but they have

not captured the sequence, or whether the TIR transposon is absent altogether (including evidence of either an empty site or excision footprint). This information is necessary to accurately interpret the evolutionary history of each Pack-TIR.

We agree with this insightful comment and revised the analyses accordingly. Now the manuscript reads as follows:

“Third, the gap-filling process induced by the fragile site (Fig. S1B) might indicate a longer waiting time between TE insertion and non-TE sequence capture compared with that obtained with the process induced by active transposon excision (Fig. S1C). Thus, TIR TEs without duplications should be present at the orthologous locus of outgroup species if duplications are specific to the focal species or lineages. We tested this hypothesis in primates due to their densely sampled phylogeny (Fig. 1E). The majority (91 out of 97, or 94%) fit the expected scenario where orthologous TIR TEs without duplications existed in the outgroup species (Table S3). Taking Pack-*Tigger1* (a *TcMariner* element) in gibbons as an example, all four outgroup primates encode orthologous *Tigger1* (Fig. 2E). The analysis of five other Pack-TIRs revealed that no orthologous TEs or TSDs (excision relics) could be detected in related primates (Fig. S4A-E). Because these five cases were from sparsely sampled lineages (*e.g.*, tarsier, Fig. 1E), the absence of homologous TEs could be explained by the lack of closely related outgroups. The last Pack-TIR was shared by primates (Fig. S4F), which suggested its ancient origin. Thus, we could not accurately infer its origination process.” (Line 252-266)

ID	Origination Model	Status of TE or Pack-TIR in the related species	Human	Chimpanzee	Bonobo	Gorilla
#203	Gap-filling	TE without duplication in Chimpanzee	+	-	-	-
#204	Gap-filling	TE without duplication in Chimpanzee	+	-	-	-
#168	Gap-filling	TE without duplication in Bonobo	-	-	-	+
#170	Gap-filling	TE without duplication in Bonobo	-	-	-	+
#160, #209	Gap-filling	TE without duplication in Chimpanzee	+	-	-	+
#127	Gap-filling	TE without duplication in Green monkey	+	+	+	+
#129	Gap-filling	TE without duplication in Orangutan	-	-	-	-

A partial snapshot of Table S3.

“Figure S4. Six Pack-TIRs completely absent in the outgroup species or shared by primates. A) Pack-TIR #269 together with the flanking 8-bp and 242-bp sequences was absent in tarsiers. B) Pack-TIR #319 together with the flanking 807-bp and 103-bp sequences was absent in squirrel monkeys. C) Pack-TIR #254 together with the flanking 1-bp and 80-bp sequences was absent in tarsiers. D) Pack-TIR #70 together with the flanking 0-bp and 8-bp sequences was absent in tarsiers. E) Pack-TIR #71 together with the flanking 457-bp and 1-bp sequences was absent in tarsiers. F) Pack-TIR #288 were present in 16 out of 18 primates. These snapshots were from the UCSC genome browser, which uses Net Track to represent genome-wide synteny. Upper-level Net is more likely orthologous, and lower-level Net might represent one-way paralogous mapping. In each case, we manually checked whether the Pack-TIR and its flanking 1000-bp sequences were shared across primates. Because outgroup species generally show

consistent patterns, we showed the net track belonging to one of the most closely phylogenetically related species for all the cases with the exception of the last case (Panel F), for which we showed tracks from multiple phylogenetically representative species.”

For the polymorphic *Ssk-FB4s*, the X-linked copy is the founding member of the whole family. We previously checked the sequences in strains without *Ssk-FB4* at this locus and wrote: “However, we did not find *FB4* insertion or TSD (excision relic) in the fly reference genome or populations including DGRP and GDL, which suggested that *FB4* is not situated in this locus.” (Line 338-340) This is a key information indicating that the generation of *Ssk-FB4* did not fit the gap-filling model, but the FoSTeST model.

The main text section on selection analyses is also very confusing as written. I think the authors intend to argue that Ssk-FB4 copies are under positive selection because nonsynonymous mutations are enriched in Ssk-FB4 functional domains, but this is not immediately apparent. Additionally, it is very unclear what the authors mean by “negative selection where 65.8% of elements are singletons,” as such an observation could also be attributed to a low transposition rate. It is also unclear to me why a higher frequency of Ssk-FB4 alleles in the population relative to other FB4 copies is necessarily indicative of positive selection. The method section is easier to understand, but the results are hard to evaluate and should be rewritten to make the rationale, logic, and methods used clearer.

Thanks for pointing out the lack of logical clarity in this section. To improve it, we added a coauthor, Dr. Hua Chen, who has experience in developing methods on detection of selection force. We then rewrote the whole section to better explain the logical flow, *e.g.*, adding an illustrative figure (Fig. S13). In addition, the skew of alleles towards low frequencies could not be explained by a low mutation rate because whether an allele could spread to a high frequency is determined by its functional effect (neutral, deleterious or beneficial) and genetic drift¹.

The current version reads as:

“Because translation does not necessarily mean function affecting fitness⁵⁴, we analyzed whether *Ssk-FB4s* were subject to positive selection by examining the substitution patterns and frequency distribution. First, because *Ssk* encodes a membrane protein³⁸, substitutions at approximately neutral sites, *i.e.*, synonymous sites, would evenly accumulate^{55,56} in intracellular, transmembrane and extracellular regions (Fig. S13A). Consistently, there is no enrichment of synonymous substitutions in three regions of *Ssk* or *Ssk-FB4s* (Fig. 5C). In contrast, substitutions at functional sites, *i.e.*, nonsynonymous sites, were overrepresented in the transmembrane domains of *Ssk* and the extracellular regions of *Ssk-FB4s*, respectively (Figs. 5C and S14B-C). Given the even distribution of synonymous substitutions, these overrepresentations suggest weaker purifying selection or stronger positive selection in the corresponding

functional regions. Second, polymorphic *FB4s* show low allele frequencies, with 65.8% as singletons (Fig. 5D, Methods). This phenomenon suggests that the spread of *FB4s* is repressed by negative selection especially considering the generally deleterious mutagenic nature of TEs⁵⁷ (Fig. S13B). In comparison, the four *Ssk-FB4s* have a higher frequency than *FB4s* (Fig. 5D). This contrast was confirmed by independent frequency data of *FB4s* (Fig. S15A). The increase in the *Ssk-FB4* frequency is either because these are less deleterious than *FB4s* or because they are subject to positive selection. We performed a test by searching for the signal of selective sweep⁵⁸, where haplotypes harboring focal mutations would have no time to accumulate many mutations if positive selection drives the rapid increase of these haplotypes (Fig. S13C). For two high-frequency copies applicable for this analysis, we indeed found the depletion of linked single nucleotide polymorphisms (SNPs, Fig. S15B-C) that could not be explained by chance ($P < 0.05$, Figs. S15D-E).” (Line 502-525)

“Figure S13. Illustrative cartoons of signatures associated with positive selection acting on *Ssk-FB4s*. A) Nonsynonymous substitutions are enriched in one particular

functional region (e.g., extracellular region) relative to the even distribution of synonymous substitutions. B) Allele frequency distribution of *FB4s* and *Ssk-FB4*. In the left panel, blue and red stars represent *FB4s*, and one *Ssk-FB4* insertion across lines, respectively. The frequency spectra of these mutations are summarized in the right panel with X and Y axis referring to the count of lines and proportion of the corresponding frequency groups. C) Mutation accumulation before and after selection. In the top panel, the red star represents *Ssk-FB4* and blue dots represent flanking mutations before (left) and after (right) selection. The bottom panel shows the corresponding nucleotide diversity across sliding windows. Under positive selection, *Ssk-FB4* will rapidly increase in frequency, and nearby linked alleles hitchhike along with it, leading to a decrease in the nucleotide diversity. This is particularly pronounced for the closely linked region due to a lack of recombination. We therefore selected synonymous substitutions with a similar allele frequency and a similar recombination rate as the neutral control to test the significance of the decrease (see also Methods).”

Finally, regarding the KO experiment, it appears that the authors only deleted the 4 base pair insertion unique to the X chromosome Ssk-FB4 copy, rather than the whole locus. It is unclear to me why the authors chose this strategy, and whether or not this 4 base pair deletion would be sufficient enough to abolish X-Ssk-FB4 function, if any. Similarly, the KO experiments are hard to interpret, as it is not clear whether the observed effects are due to the 4bp deletion or to CRISPR off targets. Thus, the authors should 1) elaborate on the rationale behind their KO design and 2) either perform a rescue experiment or temper their conclusions to say that the results of the viability assay are consistent with an effect due to loss of function of the X-Ssk-FB4 copy.

This major comment together with the minor comment (Line 680: *Is 2 generation outcrosses enough to control for background mutations? Seems like more backcrosses should have been done*) concerns our viability assay of X-linked *Ssk-FB4*, i.e., whether our KO design in the RAL-399 line is sound and whether there are potential off-targets. We appreciate these critical comments.

For KO design, we attempted to knock out the X-linked *Ssk-FB4* by a frame-disrupting indel rather than by a deletion due to the following reasons. First, indel-based KO practice is routinely performed (e.g., Cancer DepMap project²). Second, *Ssk-FB4* was situated in an intron of *kirre* and in the upstream region close to (1.0-kb) the transcriptional start site of *Syx4* (Fig. R1). Deletion of the whole *Ssk-FB4* locus (2.7 kb) may affect the local DNA folding and thus the regulation of *kirre* or *Syx4*. Finally, our 7-bp deletion (Fig. R2A) leads to a premature termination codon (PTC) and a truncated protein of only 41 amino acids (aa, 25.3% of the original length). This PTC seemingly induces nonsense mRNA decay (NMD)³ and the mRNA level is decreased to only 12.5% of the wild type level (Fig. R2B). Thus, this indel should be able to abolish the function of *Ssk-FB4*.

Figure R1. The location of X-linked *Ssk-FB4* and the flanking genes *kirre* and *Syx4*. The snapshot was from the UCSC genome browser. The thinner boxes, the thicker boxes and the intervening lines represent the untranslated regions (UTR), coding exons and introns, respectively. The blue line marks the insertion site of *Ssk-FB4*.

Regarding the concern of off-targets, the referee suggested to perform a rescue experiment or temper the conclusions. Because we attempted to render our conclusions as convincing as possible, we first tried our best to generate a rescue line, but unfortunately failed. We then performed whole genome sequencing of our two-generation outcrossed flies and examined whether the KO line harbors additional loss-of-function (LoF) mutations. Concurrently, we increased two more generations of outcrosses to further eliminate off-targets and repeated the phenotyping. Given the expected frequency of 0.25 after two generations of outcrosses, the loss of 63.8% viability in our previous experiment (Fig. R2C) could not be explained by off-targets. Consistently, sequencing data analysis indicated the absence of off-targets. However, the viability effect of *Ssk-FB4* appeared to depend on the specific environmental condition, and we could not constantly reproduce the phenotype. Since we cannot dissect the functionality of *Ssk-FB4* in a reasonable time and this manuscript is mainly about the mutational mechanism, we decided to stop the experimental trial and deleted the section to turn our tone down. Complementarily, motivated by the possibility that *Ssk-FB4s* are involved in environmental stress response, we analyzed public association data and found that *Ssk-FB4s* were always associated with stronger resistance under microbial stress (Fig. S19), which indicated their functionality. All these efforts are described with details as follows.

First, we attempted to generate a rescue strain. That is, we created a transgenic construct where the full-length X-linked *Ssk-FB4* together with the fused downstream sequence (165 bp, Fig. 4F) was amplified from RAL-399 genomic DNA and inserted into the BglIII-NheI sites of *pBac[3×P3-RFP]* vector. This vector consists of a reporter gene (RFP) facilitating the subsequent screening. We injected the transgenic vector into *Ssk-FB4*^Δ embryos at 300 ng/ul with a helper vector using standard methods, which will lead to the integration of this vector into the genome. We injected 300 embryos, collected 88 adult flies survived after the injection, and backcrossed them with *Ssk-FB4*^Δ flies. We screened ~50 progenies of each fly (a total of 4,400 flies) but failed to

detect flies with red fluorescence signal. We suspected that these *Ssk-FB4*^Δ flies are fragile under the environmental or genetic challenges and we stopped our trial.

Second, to generate our initial *Ssk-FB4* KO line, we designed a short guide RNA (sgRNA), which is uniquely mapped to X-linked *Ssk-FB4* but not the parental *Ssk* or other genomic regions. Such a design lowers the chance of off-targets. To directly test this possibility, we performed whole genome sequencing and mutation calling. That is, by comparing our two-generation outcrossed *Ssk-FB4*^Δ and RAL-399 wild type, we identified 7,537 SNPs and indels only present in the KO line (Table R1). We also searched for structural variations (SVs), but did not detect any SVs private to *Ssk-FB4*^Δ. Out of 7,537 mutations, we only found one potential LoF mutation (deletion of “C” at chrX: 3,066,813) apart from the aforementioned indel disrupting *Ssk-FB4* (Table R2). This 1-bp deletion is very unlikely to be a LoF mutation caused by off-target effects of our CRISPR/Cas9 practice due to the following reasons: 1) this heterozygous deletion situates close to the 3’ end of *CG4116* and only removed 8 aa (3.4% of the protein); 2) *CG4116* represents an uncharacterized gene model with only a predicted protein lacking any known domains or motifs and it is unexpressed or lowly expressed in all FlyBase larva or adult samples except larva L2 whole body sample; and 3) our sgRNA does not share similarity with the region harboring this deletion and thus should not be able to induce mutation in this locus.

Table R1. Mutations detected in *Ssk-FB4*^Δ but not in RAL-399 wild type.

	SNPs and indels (GATK)	SVs (LUMPY)	SVs (CNVnator)
All Ssk-FB4 ^Δ -specific mutations	7,537	0	0
LoF mutations	2	0	0

The NCBI accession numbers for *Ssk-FB4*^Δ and wild type resequencing data are SRR14026713 and SRR13154263, respectively. We mapped the reads to the reference genome via BWA v0.7.17-r1188⁴. We used GATK v3.7 to detect SNPs and indels, according to the GATK best practices⁵, and LUMPY v0.2.13⁶ or CNVnator v0.3.3⁷ to call SVs. The functional consequences of SNPs and indels were annotated by SnpEff v5.0c⁸. Unless otherwise stated, all software were run with their default parameters⁹.

Table R2. Two candidate LoF mutations.

Locus	Mutation	Allele frequency
X-linked Ssk-FB4	GCGGCACG->G	1
ChrX:3,066,813 (CG4116)	GC->G	0.5

Allele frequency is estimated by proportion of reads harboring the focal mutation.

Finally, regarding the number of outcross generations, we previously did only two generations due to the following reasons: 1) we did not have much manpower in the early last year due to the pandemic; and 2) with two generations of outcrosses, the frequency of off-target mutations (if exist) would drop to 0.25 and it would continue to drop in the subsequent maintenance if the mutations were deleterious. Actually, given the frequency of 0.25, the loss of 63.8% viability in our previous experiment (Fig. R2C) could not be explained by the off-target. Nonetheless, with the increase of manpower during the revision stage (late last year), we controlled for off-target mutations by adding another two generations of outcrosses (Fig. R2D). We then repeated phenotyping after raising flies in small vials with fresh food (which was called normal condition). Surprisingly, there was no significant difference of the survival rate between wild type and *Ssk-FB4*^Δ flies this time (Fig. R2C). We noticed that the survival rate of *Ssk-FB4*^Δ flies was more variable between replicates than wild type (standard deviation, 0.13 vs. 0.06), and suspected that the functionality of *Ssk-FB4* depended on the specific environmental condition. We thus recalled that due to lack of manpower early last year we unintentionally raised the two-generation outcrossed flies in large bottles to minimize the frequency of transferring flies to fresh media. However, the food in large bottles may become stale by this way, and contain bacteria, fungi, or even mites (which was called “overstressed” condition). We tried to mimic such an “overstressed” condition and carried out the viability assay for the third time by raising flies in large bottles. *Ssk-FB4*^Δ flies are still not different from the wild type in terms of mean viability (Fig. R2C) possibly because we could not make the environment exactly the same as that in last year.

Figure R2. Survival rate assay. A) Knockout strategy and genotyping of *Ssk-FB4*^Δ. The 7-bp deletion was confirmed by Sanger sequencing. The gene schema follows Fig. 4F. The red lines mark the nucleotide differences between *Ssk-FB4* and *Ssk* in the RAL-

399 line with sgRNA (underlined) harboring one differentiating “A” (shaded in red). It should be noted that X-linked *Ssk-FB4* is the only copy in the RAL-399 line (Table S5). B) qRT-PCR result showing expression change of *Ssk* and *Ssk-FB4* in the midgut. The bars of standard errors were based on two biological replicates. Wilcoxon rank test was used to calculate *P*-value for fold changes of *Ssk-FB4*. C) Survival rate of wild type and *Ssk-FB4*^Δ. The rate was calculated as the proportion of flies viable from eggs through to adults. The bars of standard errors were calculated with four, eight or seven biological replicates, respectively. Dots show the observed survival rate. D) Schema of additional two-generation outcrosses and survival rate calculation. In total, we performed four-generation outcrosses.

Taken together, *Ssk-FB4*'s function appears to be associated with specific environmental stress, which could not be pinpointed via the current experiments. We are then interested in what exact environmental pressure could be. Because *Ssk* is known to be involved in the intestinal epithelial barrier, gut homeostasis and immune response, we did a survey of studies on related traits, and found only four published genome-wide association study (GWAS) datasets with enough statistical power. Interestingly, all datasets consistently show that the presence of *Ssk-FB4s* were associated with stronger resistance against viral, bacterial and fungal infection. The effect size of *Ssk-FB4s* is variable for different pathogens where the viability of flies harboring *Ssk-FB4s* was significantly higher in two fungal infection experiments and marginally higher for an additional bacterial infection experiment (Fig. S19). The resistance variability of *Ssk-FB4s* together with inconstancy of environmental microbe composition may explain why we could not reproduce our viability phenotype. Since we have never cultivated microorganisms in our lab and it may take some time for us to set up the culture condition especially for fungi, we did not perform infection assay in *Ssk-FB4*^Δ and wild type. As this manuscript is mainly about the mutational mechanism of TIR TE-induced duplication, we decided to delete the KO section and toned down our statements accordingly. Complementarily, we added the analysis results of GWAS datasets. We plan to collaborate with labs who has expertise in fly immune response to finally figure out how *Ssk-FB4s* exert its role in the future.

We revised the Discussion and Methods accordingly:

“The side function of *Ssk* or the major function of *Ssk-FB4s* is intriguing. Although *Ssk* is known to be involved in the intestinal epithelial barrier, gut homeostasis and immune response^{59,69,70}, no studies have pinpointed the key residues underlying these processes; Therefore, we could not infer the function of *Ssk-FB4s* based on amino acid differences relative to *Ssk*. Nonetheless, *Ssk-FB4s* are likely involved in similar processes as *Ssk* given their similar expressional control. To infer the functionality of *Ssk-FB4s*, we analyzed public genome-wide association study (GWAS) datasets (Methods) from flies subjected to microbial infection and whose gut homeostasis or immune response was challenged. We found that the presence of *Ssk-FB4s* was always associated with stronger resistance against viral, bacterial and fungal infection in all datasets⁷¹⁻⁷³: the

comparison reaches marginal significance in one set with female flies subjected to bacterial infection ($P = 0.06$, Fig. S19), and significance in another two sets with flies subjected to fungal infection ($P = 0.002$ and $P = 0.02$, respectively). Whether and how they shape this phenotype warrants further studies.” (Line 619-632)

“**Figure S19. Survival status of DGRP lines infected with pathogens.** A) DGRP lines infected with the WNV subtype Kunjin virus. Hazard ratio refers to the death rate of the infected group compared with control group, and $-\log(\text{hazard ratio})$ indicates the survival rate. B) DGRP lines infected with the bacterium *Pseudomonas entomophila*. C) DGRP lines infected with the bacterium *Pseudomonas aeruginosa* Pa14. D) Viability of DGRP lines infected with the fungus *Metarhizium anisopliae* Ma549. For Panel C and D, the viability was quantified as LT50, *i.e.*, the days needed for half the flies to die.”

“GWAS data analyses

Since *Ssk-FB4s* could be involved in gut homeostasis or immune response, we performed a literature survey about public GWAS studies focusing on these traits in DGRP lines and found 13 studies (Table S11). Considering the moderate population frequency of *Ssk-FB4s*, we focused on only four datasets with sufficient statistical power (more than 80 lines). These datasets examined the viability of different DGRP lines upon the infection of the West Nile virus subtype Kunjin (WNV-Kun)⁷¹, the bacterium *Pseudomonas entomophila*⁷² or *Pseudomonas aeruginosa* Pa14⁷³ and the fungus *Metarhizium anisopliae* Ma549⁷³. We divided the DGRP lines into two groups based on the presence or absence of *Ssk-FB4s* and compared the viability between these two groups.” (Line 874-884)

Overall, I think that, once these issues with the writing and experimental detail are resolved, the paper will be a useful contribution to the field.

Again, we appreciate your constructive review.

Specific Comments

- *Line 51: “renovations” is an unusual word choice; typically “innovations” is used*

We agree and made the revision accordingly.

- *Line 54-55: Here and elsewhere in the paper the authors argue that the mechanism of gene duplication affects their functional trajectories, which in principle is probably correct, but I don’t think statements like “duplicates generated by TEs are more likely neofunctionalized due to various mechanisms” is accurate*

Thanks for pointing out this issue. We have now modified the sentence to make our statement more precise:

“The mechanism responsible for the generation of duplicates affects their evolutionary trajectories^{7,8}. Duplicates generated by TEs are more likely to evolve new structures or functions due to the formation of chimeric transcripts or changes in the regulatory context⁹⁻¹¹. Therefore, the mechanism through which TEs mediate duplications is of broad interest.” (Line 57-61)

- *Line 71: Here the authors should be clear what they mean by “spatial proximity” – ie: linear or 3D proximity?*

We have now clarified:

“During the repair, the template could switch from the sister strand to adjacent external sequences (called fillers) in 3D proximity, leading to the capture of fillers^{21,25}.” (Line 75-77)

- *Lines 157-159: While it is certainly possible that a linear correlation between the # of Pack-TIRs and the total number of TIRs in the genome could suggest a transposition independent mechanism, it could equally just be due to the fact that the presence of more TE copies increases the overall opportunity to generate a Pack-TIR.*

We agree. We have now toned down our writing and added a new analysis supporting the inactivity of TEs.

“Because different TE families could exhibit different extents of transposition activity, the linear relationship between the number of Pack-TIRs and consensus TIR TEs appears to suggest a transposition-independent duplication process. More direct evidence came from the inactivity of many consensus TEs^{35,36}. In actuality, the consensus sequences of TEs tend to be short, with a median length of only 374 base pairs (bp). Moreover, individual TE copies are different from the consensus by a median

divergence value of 18.3% (Table S1). These two patterns suggest that TEs likely represent degenerate ancient relics.” (Line 162-169)

- *Line 177: “...21 cases merged from orthologs” It is unclear what the authors mean by “merged from orthologs” both here and in the methods*

To clarify, we changed the Results:

“As an example, the human genome harbors 33 Pack-TIRs, including 12 cases directly called by our pipeline (Fig. 1A) and 21 cases called at orthologous loci in nonhuman primates but exhibiting slightly lower alignment quality in humans (not passing the cutoffs, Tables S3-4, Methods).” (Line 185-189)

We also modified the Discussion:

“Specifically, our survey identified only 12 Pack-TIRs in humans (Fig. 1A). However, manual curation showed that we missed 21 cases identified in other primates (Fig. 1E), mainly because orthologous cases in humans are slightly below the cutoffs (*e.g.*, identity cutoff with parental copies).” (Line 584-587)

- *Line 182: Here and throughout the paper the authors use “dominance” when I think it would be better to use “common” or “most common” as dominance has a very particular genetics definition*

We agree and modified all phrases where “dominance” or “dominant” were used.

“Most (323 or 87.2%) Pack-TIRs were associated with the top two most common superfamilies, *hAT* (208) and *TcMariner* (115) (Table S1), which jointly contributed 81.8% of the TIR TE content in the 100 species (Table S2).” (Line 136-139)

“In summary, the linear relationships among the numbers of Pack-TIRs and consensus TEs, the prevalence of single-copy Pack-TIRs, and the origination timing of Pack-TIRs in primates jointly suggest that these were mainly generated via a transposition-independent mechanism.” (Line 191-194)

“Consistently, for 281 unique Pack-TIRs in the reference genomes (after controlling for redundant ones in primates and multicopy cases, Table S1), TIR TEs capturing sequences from the same chromosome (intrachromosomal duplication) were widespread in animals (59.8%, Fig. 2A).” (Line 200-203)

“We used the major isoform according to expression sequence tag data from the UCSC genome browser³².” (Line 722-724)

- *Line 236: What do the authors mean by “polarize the evolutionary order”?*

We previously referred to determine the sequential order of evolutionary events including transposition (insertion of transposons) and duplication (acquiring non-TE sequences). With such information, we could infer whether duplication occurs after or at the same time of transposition. Together with the comment of Referee 1, we expanded this section as follows:

“The majority (91 out of 97, or 94%) fit the expected scenario where orthologous TIR TEs without duplications existed in the outgroup species (Table S3). Taking Pack-*Tigger1* (a *TcMariner* element) in gibbons as an example, all four outgroup primates encode orthologous *Tigger1* (Fig. 2E). The analysis of five other Pack-TIRs revealed that no orthologous TEs or TSDs (excision relics) could be detected in related primates (Fig. S4A-E). Because these five cases were from sparsely sampled lineages (*e.g.*, tarsier, Fig. 1E), the absence of homologous TEs could be explained by the lack of closely related outgroups. The last Pack-TIR was shared by primates (Fig. S4F), which suggested its ancient origin. Thus, we could not accurately infer its origination process.” (Line 257-266)

ID	Origination Model	Status of TE or Pack-TIR in the related species	Human	Chimpanzee	Bonobo	Gorilla
#203	Gap filling	TE without duplication in Chimpanzee	+	-	-	-
#204	Gap filling	TE without duplication in Chimpanzee	+	-	-	-
#168	Gap filling	TE without duplication in Bonobo	-	-	-	+
#170	Gap filling	TE without duplication in Bonobo	-	-	-	+
#160, #209	Gap filling	TE without duplication in Chimpanzee	+	-	-	+
#127	Gap filling	TE without duplication in Green monkey	+	+	+	+
#129	Gap filling	TE without duplication in Orangutan	-	-	-	-

A partial snapshot of Table S3.

“Figure S4. Six Pack-TIRs completely absent in the outgroup species or shared by primates. A) Pack-TIR #269 together with the flanking 8-bp and 242-bp sequences was absent in tarsiers. B) Pack-TIR #319 together with the flanking 807-bp and 103-bp sequences was absent in squirrel monkeys. C) Pack-TIR #254 together with the flanking 1-bp and 80-bp sequences was absent in tarsiers. D) Pack-TIR #70 together with the flanking 0-bp and 8-bp sequences was absent in tarsiers. E) Pack-TIR #71 together with the flanking 457-bp and 1-bp sequences was absent in tarsiers. F) Pack-TIR #288 were present in 16 out of 18 primates. These snapshots were from the UCSC genome browser, which uses Net Track to represent genome-wide synteny. Upper-level Net is more likely orthologous, and lower-level Net might represent one-way paralogous mapping. In each case, we manually checked whether the Pack-TIR and its flanking 1000-bp sequences were shared across primates. Because outgroup species generally show

consistent patterns, we showed the net track belonging to one of the most closely phylogenetically related species for all the cases with the exception of the last case (Panel F), for which we showed tracks from multiple phylogenetically representative species.”

• *Line 252-254: Do the authors observe an empty site in the African clawed frog or is it just absent from the assembly?*

There is no signature of an empty site because the whole orthologous locus together with flanking regions is absent. However, this pattern could be caused by the fact that it lacks closely related outgroups. So, to clarify, we modified this section as:

“Contrary to most single-copy Pack-TIRs in primates, we could not find orthologous TIR TEs for all four Pack-TIRs. Instead, the syntenic regions do not encode Pack-TIRs together with nearby flanking regions. Taking Pack-*hAT-N8_XT* (Fig. 1D) as an example, all three derived Pack-TIRs or *hAT-N8_XT* without duplication were absent in the orthologous loci of the outgroup, the African clawed frogs (Fig. S5), which diverged from the western clawed frogs 57 Mya³⁷. Thus, either the multicopy cases do not fit the gap-filling model or the outgroup species is too divergent to provide sequence information indicating the origination process of these Pack-TIRs.” (Line 283-291)

• *Line 320-322: The conclusion presented here is too strong for the data – the scenario the authors propose is only one possibility*

We agree and revised as follows:

“Taken together, the absences of orthologous TEs without duplication across multicopy Pack-TIRs in frogs, alligators and flies suggest that multicopy Pack-TIRs were likely generated through the FoSTeST process.” (Line 364-366)

• *Lines 335-347: For the transcriptome analysis, the authors should elaborate from which promoter the gene fragments are transcribed from. The TE? An upstream/downstream gene? A cryptic promoter?*

We agree. We previously mainly focused on *Ssk-FB4s* and showed that their transcription profiles should be determined by the regulatory context of both *Ssk* and *FB4* (Fig. 4G). We now clarify cases in humans and worms as follows:

“Note: the transcription of these eight cases is likely driven by promoters of host genes because their transcriptional orientation is the same as that of the host genes in seven cases and their expression breadth is identical to that of the host genes in six cases (*e.g.*, *GANQ-MER5A*, Fig. 4C). The last two cases are relatively long: a retrogene *ARF6*⁵⁰ was almost completely captured into the *MER20* transposon, but this duplication could also reflect a recurrent retroposition event (Fig. S9B, Methods); the 3' UTR of *SP1* was

duplicated with *MER20* as a noncoding RNA (ncRNA, Fig. S9C). In both cases, their expression appears shaped by flanking host regions and/or *MER20* transposons with known promoter activity⁵¹.” (Line 422-430).

“Note: because this Pack-TIR was inserted into the upstream region of *rpb-4* with the intervening gene *mtch-1* on the antisense strand, its transcription appears driven by the 5’ flanking transposon.” (Line 440-442)

- *Lines 427-428: The authors should be clear “30% or more” Pack-TIRs are transcribed in these three species, and that further studies are necessary to determine if this is generalizable across all animal Pack-TIRs*

We agree and revised accordingly.

“Overall, the profiling of limited transcriptome diversity across three species revealed that 30% or more Pack-TIRs are transcribed as chimeric transcripts. Whether this is generalizable across all animal Pack-TIRs warrants further analysis.” (Line 465-467)

- *Line 466: “Since translation is not equal to function affecting fitness” is grammatically awkward*

Please see the next response.

- *Line 467: The authors should be precise about what they mean by “evolve under natural selection”*

As in response to earlier major comments regarding “function” and methods detecting selection (Page 18), we revised this section as:

“Because translation does not necessarily mean function affecting fitness⁵⁴, we analyzed whether *Ssk-FB4s* were subject to positive selection by examining the substitution patterns and frequency distribution.” (Line 502-504)

- *Lines 492-495: It is not clear to me why an absence of interaction between X-Ssk-FB4 protein and mesh/Tsp2A would be expected to affect the expression of these two genes. The authors should elaborate more on the mesh/Tsp2A angle so that the reader can better understand how their data fits into known roles for these genes*

Thank you for pointing out this issue. We now explain our rationale in more details:

“Ssk mainly exerts its function by forming a protein complex with Mesh and Tsp2A at smooth septate junctions (sSJs)^{52,59}. Possibly due to the stoichiometric balance between protein complex members⁶⁰, these sSJ proteins appear to be subject to coregulation and thus maintain a dosage balance. For example, in response to environmental changes,

Ssk and *Mesh* are consistently up- or downregulated⁶¹. Therefore, we analyzed whether *Ssk-FB4s* are coregulated with *Mesh* or *Tsp2A*. We first found that the expression of *mesh* or *Tsp2A* was not upregulated despite the extra dosage of *Ssk-FB4s* (Figs. 5E and S16). Moreover, after co-immunoprecipitation with an antibody against *Mesh*, we quantified the protein intensity of *Ssk-FB4* and *Ssk*. Despite the 500% abundance of X-linked *Ssk-FB4* relative to that of *Ssk* in the fly midgut, the former abundance dropped to 70% of the latter in the immunoprecipitates (Fig. 5F). Thus, *Ssk-FB4s* appear to not be involved in the protein complex of *Ssk/Mesh/Tsp2A* or only weakly interact with this complex. Together with the contrastive substitution pattern (Fig. 5C), *Ssk-FB4s* and *Ssk* very likely have different functions.” (Line 526-539)

• *Line 566: It is not clear what the authors mean by “conferring side function”*

The related conceptual framework is proposed by the IAD model and we have clarified accordingly:

“With increases in the copy number, they likely evolve under the adaptive radiation model or innovation, amplification and divergence (IAD) model^{7,68}. In other words, *Ssk* exhibits trace-level beneficial side activity (innovation) but cannot be optimized due to the antagonistic constraint of its main function. With amplification to increase the mutational targets, positive selection specifically targets the side function of the derived copies and leads to divergence. Consistently, substitutions mainly occur in transmembrane (improving the main function) and extracellular (improving the side function) regions of *Ssk* and *Ssk-FB4s*, respectively; and *Ssk-FB4s* weakly interact with *Mesh* or *Tsp2A*.” (Line 606-614)

• *Lines 588-591: The authors limit their pipeline to only identify short genic insertions – did the authors ever find larger insertions?*

We clarified as follows:

“We identified young Pack-TIRs (Fig. S20A) with unambiguous parental copies by modifying a previous method⁶⁷. Specifically, the UCSC genome browser database hosts 106 animal species (<https://genome.ucsc.edu>, December 2016), in which TEs have been annotated in 100 genomes. After downloading the genome sequences and TE annotations for these species, we searched for sequences flanked by two TIR TEs belonging to the same element where the internal sequences were longer than 100 bp and shorter than 5,000 bp by following routine practices^{67,88}. Because Pack-MULEs are short (~300 bp), 5,000 bp is sufficient to cover most *bona fide* Pack-TIRs. Taking humans as an example, the largest case is *ARF6* (3,585 bp). We also tried an alternative cutoff of 10,000 bp in humans and could not find additional Pack-TIRs.” (Line 657-666)

- Lines 622-629: Overall the method to simulate the homology breakpoints is not clear – the authors should elaborate on this point

We added an illustrative panel (Fig. S3A) and revised the Methods accordingly.

“For 32 cases, we took the parental copy together with its 5’ and 3’ flanking regions (half length of the parental copy) as the template (Fig. S3A). We randomly selected 100 fragments from the parental locus with the same length as the captured sequence of the corresponding Pack-TIR and randomly selected the switching points in the consensus TIR TEs. We recorded the size of the microhomology at the breakpoint of interest and counted the proportion of replicates with equal or longer microhomology as the empirical P -value. The microhomology length distribution of 100 replicates is depicted as boxplots in Fig. S3B. Taking case #02 as an example, only six replicates showed equal or longer microhomology than the length of observed microhomology, and we thus labeled this example as ‘ $P = 0.06$.’” (Line 697-706)

“Figure S3A. Simulation process testing the length of the microhomology at the left breakpoint. The figure convention follows that of Fig. 2E with the exception that the flanking regions of the parental copy are shown in light blue. The parental copy and its flanking regions (50% length of the parental copy) were used as the template. Switching points in the consensus TIR TE and parental copy were randomly selected. After constructing 100 pseudo Pack-TIRs, we recorded the size of the microhomology (marked in orange) at the left breakpoint. Four replicates are shown as examples, microhomology was detected for two replicates, and none of these was as long as the observed microhomology.”

- Line 680: Is 2 generation outcrosses enough to control for background mutations? Seems like more backcrosses should have been done

This specific comment is related to the final major comment. We therefore address them together on Pages 21-26.

• *Lines 728-732: The authors should elaborate a bit more on how they performed de-novo transcriptome assembly, including which reads were included in the model – the text makes it seem that only the Pack-TIR reads were included which, if the case, could lead the authors to miss the overall exon/intron structure of transcripts containing Pack-TIRs*

We apologize that we did not clearly explain this point. We actually used reads mapping to flanking regions of Pack-TIRs as well. We modified the text accordingly.

“We retained only uniquely mapped reads with the criteria NH:i:1 in the BAM files¹¹⁰. We conservatively required at least five reads to define the expressed Pack-TIRs. We retrieved reads mapped to the Pack-TIRs and 100-kb flanking regions and performed *de novo* assembly using Trinity v2.6.5¹¹¹ with the parameter ‘--SS_lib_type RF’ because the RNA-seq data are strand-specific. As a result, we should be able to infer the overall exon/intron structure of transcripts containing Pack-TIRs. We further aligned the assembled contigs to the genome via BLAT⁸⁹.” (Line 795-801)

Reviewer #3 (Remarks to the Author):

Tan and colleagues present comprehensive analysis of DNA-transposon-mediated gene duplications in metazoan. This is very interesting analysis and worth consideration by Nature Communication. However, there are several issues that should be resolved before the manuscript can be published.

We appreciate the encouraging comments of the referee and addressed the issues as follows.

Firstly, the title is a bit misleading because most of presented evidence are pointing to pieces of genomic DNA, not the real genes that were duplicated with aid of transposon action. I suggest that the title of the manuscript should be modified accordingly to better reflect the main findings.

We previously used this title because most DNA TEs captured exonic or intronic sequences and generated new gene structures or new genes. Now, together with new analyses further supporting transposition-independent gap-filling process motivated by the suggestion of Referee 1 (Page 5-9), we changed the title to “DNA transposons mediate duplications via transposition-independent and -dependent mechanisms in metazoans”.

Another issue is functional analysis of duplicated genes and Ssk-FB4s in particular. Although, knock out experiments and evolutionary analysis showed that Ssk-FB4s gene is essential and its function is different from the source gene (Ssk), the authors didn't demonstrate what the new ruction could be. I would very much appreciate the discussion, even speculative one, on the potential function of Ssk-FB4s. Additionally, a comprehensive comparison of proteins coded by Ssk-FB4 and Ssk genes would enhance the discussion of this interesting gene.

We agree and thus attempted new analyses of *Ssk-FB4s* and *Ssk* in addition to our previous analyses (e.g., substitution pattern in Fig. 5C). Because computational prediction could generate false positives while not providing novel insights on functional difference between these highly similar paralogs, we decided to perform structural analyses, association analyses and a literature survey. Specifically, we predicted the protein structure with the I-TASSER server, which was “ranked as the No. 1 server for protein structure prediction” (<https://zhanglab.ccmb.med.umich.edu/I-TASSER/>). However, due to lack of homologous template in the structural database, the prediction result of *Ssk-FB4* was not reliable (TM-score < 0.5). Thus, we instead tried to infer the function of *Ssk-FB4s* based on the knowledge of *Ssk* and public association studies. We revised the Discussion and Methods accordingly.

“The side function of *Ssk* or the major function of *Ssk-FB4s* is intriguing. Although *Ssk*

is known to be involved in the intestinal epithelial barrier, gut homeostasis and immune response^{59,69,70}, no studies have pinpointed the key residues underlying these processes; Therefore, we could not infer the function of *Ssk-FB4s* based on amino acid differences relative to *Ssk*. Nonetheless, *Ssk-FB4s* are likely involved in similar processes as *Ssk* given their similar expressional control. To infer the functionality of *Ssk-FB4s*, we analyzed public genome-wide association study (GWAS) datasets (Methods) from flies subjected to microbial infection and whose gut homeostasis or immune response was challenged. We found that the presence of *Ssk-FB4s* was always associated with stronger resistance against viral, bacterial and fungal infection in all datasets⁷¹⁻⁷³: the comparison reaches marginal significance in one set with female flies subjected to bacterial infection ($P = 0.06$, Fig. S19), and significance in another two sets with flies subjected to fungal infection ($P = 0.002$ and $P = 0.02$, respectively). Whether and how they shape this phenotype warrants further studies.” (Line 619-632)

“**Figure S19. Survival status of DGRP lines infected with pathogens.** A) DGRP lines infected with the WNV subtype Kunjin virus. Hazard ratio refers to the death rate of the infected group compared with control group, and $-\log(\text{hazard ratio})$ indicates the survival rate. B) DGRP lines infected with the bacterium *Pseudomonas entomophila*. C) DGRP lines infected with the bacterium *Pseudomonas aeruginosa* Pa14. D) Viability of DGRP lines infected with the fungus *Metarhizium anisopliae* Ma549. For Panel C and D, the viability was quantified as LT50, *i.e.*, the days needed for half the flies to die.”

“GWAS data analyses

Since *Ssk-FB4s* could be involved in gut homeostasis or immune response, we performed a literature survey about public GWAS studies focusing on these traits in DGRP lines and found 13 studies (Table S11). Considering the moderate population frequency of *Ssk-FB4s*, we focused on only four datasets with sufficient statistical power (more than 80 lines). These datasets examined the viability of different DGRP lines upon the infection of the West Nile virus subtype Kunjin (WNV-Kun)⁷¹, the bacterium *Pseudomonas entomophila*⁷² or *Pseudomonas aeruginosa* Pa14⁷³ and the

fungus *Metarhizium anisopliae* Ma549⁷³. We divided the DGRP lines into two groups based on the presence or absence of *Ssk-FB4s* and compared the viability between these two groups.” (Line 874-884)

Finally, I would love to see discussion/comparison of the described here phenomenon to DNA transduction mediated by L1 and SVA elements described previously by several groups.

Thanks for this comment, which has encouraged us to frame our discovery in a broader context. Indeed, transductions mediated by L1s or SVAs share common features with duplications mediated by Pack-TIRs. We accordingly revised the Introduction and Discussion. Note, transposons are remarkably diversified in terms of specific transposition mechanism and species distribution. Thus, we herein limited our discussion to their commonality rather than difference.

“Studies, including ours, have shown that long terminal repeats (LTRs) and non-LTR retrotransposons, such as L1 or SVA elements, mediate the retroduplication of host messenger RNAs (mRNAs) in animals and plants^{10,12-15}.” (Line 62-64)

“Finally, our results together with those obtained in prior studies^{10,12,14,15} show that both DNA transposons and retrotransposons catalyze duplications in animals. Despite their different transposition mechanisms and distribution across species, DNA transposons and retrotransposons share four features. First, the end bypass model (Fig. S1A) can apply to the 3' transduction mediated by TIR TEs²⁰, *Helitrons*⁸², L1s^{15,83} and SVAs¹⁴. Second, template switching during the gap-filling (Fig. S1B-C) or FoSTeST model (Fig. 6) is also commonly observed in the duplication process mediated by *Helitrons*⁸⁴, L1s⁸⁵ and LTR retrotransposons¹⁰. Third, duplicates flanked by TEs could act as pseudo TEs and amplify via further transpositions, such as the *Ssk-FB4* family described here, the *AMAC* family mediated by SVA^{14,86} and the *CG17604_r* family mediated by LTR retrotransposon¹⁰. Fourth, chimerism between duplications, flanking TEs or insertion sites is widespread not only for Pack-TIRs but also for *Helitrons*¹⁶, LTRs¹⁰, L1s^{85,87} and SVAs¹⁴. All of these features confer TEs with a strong capability of shuffling genetic materials and endorse TEs as a vibrant force in shaping the evolution of new genes and new gene structures in animals.” (Line 640-654)

Minor pints and clarifications.

1. The authors should pay better attention to detail description of the figures and avoid discrepancies in the presented results. For instance, in Figure 1 A we are presented with information on distribution of Pack-TIRs in 55 animals with some of the species having no Pack-TIRs detected. However, in the beginning of the paragraph we can read that 100 animal genomes were analyzed and Pack-TIRs were found within 55 genomes (line 109 and 120, respectively). Additionally, in the same figure legend we can read

“To simplify, 18 primates and 12 *Drosophila* species were represented by human and *D. melanogaster*, respectively.” I understand the authors intention but why not to label the respected phylogenetic tree leaves with “primates” and “*Drosophila*?” Interestingly, there is no *D. melanogaster* label but fruitfully that most likely represents *Drosophila* lineage.

Thanks for this specific and helpful comment. We followed your advice and changed “human” to “primates”, and “fruitfly” to “*Drosophila*”, respectively. The total numbers of Pack-TIRs in both lineages were changed accordingly. The updated Fig. 1A together with the legend is as follows:

“Figure 1A. Distribution of Pack-TIRs in 100 animals. The number of Pack-TIRs (if any) is shown close to the taxon name, and the size of the pink background is proportional to the number. Note: the primate order (18 species) and the *Drosophila* genus (11 species) are collapsed, and the total number of Pack-TIRs (if present) is shown.”

What colored numbers in Figure3C represent?

They are DGRP lines encoding *Ssk-FB4s*. We modified both the figure and the legend to clarify what they represent.

“Figure 3C. A phylogenetic tree of Ssk-FB4s and Ssk. ... Ssk-FB4s sequenced in a set of *Drosophila* lines served as ingroups with the colored IDs referring to individual DGRP lines (e.g., RAL-379)...”

2. “Microhomology” term. Homology is an evolutionary term that indicates common ancestry of two feature, e.g. piece of DNA. However, the authors use this term to refer to two or more short similar sequence without implying their common evolutionary ancestry. I strongly suggest to avoid microhomology term throughout the manuscript unless the authors would like indicate their common ancestry, which I believe is not the case.

Thanks for this insightful comment. Indeed, we did not intend to imply “their common ancestry”. Microhomology (or microsimilarity) means a short similar sequence shared by the two templates. Strictly speaking, the term “microhomology” is conceptually wrong. However, in the field of mutational mechanisms, microhomology rather than microsimilarity is ubiquitously used, e.g., microhomology-mediated end joining (MMEJ)¹⁰ and microhomology-mediated break-induced replication (MMBIR)¹¹. Thus, it should be better to follow this convention used in the field. To remind readers, we added a sentence to define microhomology.

“Second, template switching in the gap-filling model is expected to occur under the guidance of microsimilarity (a short, similar sequence shared by the two templates; also called microhomology, which was used hereafter) shared by TEs and fillers²⁵.” (Line 241-243)

3. Lines 448-450. “The lower nonsynonymous data is based on the whole *Drosophila* genus to increase the statistical power (Fig. S12E), while the synonymous data is only based on melanogaster subgroup to avoid the saturation of synonymous changes.” I’m not sure if this approach is methodologically acceptable. Altering data sets to get better statistical support is a dangerous operation as one can easily loose objectivity of data analysis.

We deeply appreciate that the referee pointed out this methodological problem. The reason that we performed the analysis in this way is because *Ssk* is highly and consistently constrained across almost all evolutionary branches (Fig. S14E). The median number of nonsynonymous substitutions is zero. To increase the statistical power, we had to use the phylogenetic tree of the whole genus. However, in this scenario, synonymous substitutions are saturated. We thus only used the phylogeny of the *melanogaster* subgroup given the consistent evolution pattern across all branches.

We have now used the whole genus data for both synonymous and nonsynonymous substitutions in *Ssk*. To address the saturation issue, we replaced the previous Fisher’s Exact Test with a proportional test and reproduced the previous observations. We updated the figure and legend accordingly.

C

		Intracellular	Transmembrane	Extracellular	P
Ssk-FB4	Nonsynonymous	0 (0/0.12)	4.2 (0.40/0.59)	6.3 (0.60/0.29)*	0.03
	Synonymous	0 (0/0.16)	5.9 (0.86/0.59)	1 (0.15/0.25)*	0.58
Ssk	Nonsynonymous	2.1 (0.069/0.14)	25.3 (0.83/0.57)*	3.1 (0.10/0.30)	0.0028
	Synonymous	74.4 (0.16/0.11)	297.8 (0.62/0.63)*	108 (0.23/0.26)	0.68

“**Figure 5C. Numbers of nonsynonymous and synonymous changes across three functional regions in *Ssk-FB4s* and *Ssk*.** The upper polymorphism data are based on the codon-level alignment of *Ssk-FB4s* (Fig. S14A). The lower divergence data of *Ssk* are based on the whole *Drosophila* genus (Fig. S14E). Each cell is shown as the ‘number of changed sites (observed/expected proportion of changed sites)’. **P*-values were calculated for the regions of interest with the proportional test.”

We would like to thank the editor and all 3 reviewers again for their positive evaluation and constructive comments, which have helped us to significantly improve our manuscript.

References

1. Graur, D. & Li, W.-H. *Fundamentals of Molecular Evolution 2nd ed. (Chapter 2: Dynamics of Genes in Populations)*, (Sinauer Associates, Sunderland (MA), 2000).
2. Tsherniak, A. *et al.* Defining a Cancer Dependency Map. *Cell* **170**, 564-576.e16 (2017).
3. Yang, H. *et al.* Expression Profile and Gene Age Jointly Shaped the Genome-Wide Distribution of Premature Termination Codons in a *Drosophila melanogaster* Population. *Molecular Biology and Evolution* **32**, 216-228 (2014).
4. Li, H. & Durbin, R. Fast and accurate long-read alignment with Burrows–Wheeler transform. *Bioinformatics* **26**, 589-595 (2010).
5. McKenna, A. *et al.* The Genome Analysis Toolkit: A MapReduce framework for

- analyzing next-generation DNA sequencing data. *Genome Research* **20**, 1297-1303 (2010).
6. Layer, R.M., Chiang, C., Quinlan, A.R. & Hall, I.M. LUMPY: a probabilistic framework for structural variant discovery. *Genome Biology* **15**, R84 (2014).
 7. Abyzov, A., Urban, A.E., Snyder, M. & Gerstein, M. CNVnator: An approach to discover, genotype, and characterize typical and atypical CNVs from family and population genome sequencing. *Genome Research* **21**, 974-984 (2011).
 8. Cingolani, P. *et al.* A program for annotating and predicting the effects of single nucleotide polymorphisms, SnpEff. *Fly* **6**, 80-92 (2012).
 9. Van der Auwera, G.A. *et al.* From FastQ Data to High-Confidence Variant Calls: The Genome Analysis Toolkit Best Practices Pipeline. *Current Protocols in Bioinformatics* **43**, 11.10.1-11.10.33 (2013).
 10. McVey, M. & Lee, S.E. MMEJ repair of double-strand breaks (director's cut): deleted sequences and alternative endings. *Trends in Genetics* **24**, 529-538 (2008).
 11. Zhang, F. *et al.* The DNA replication FoSTeS/MMBIR mechanism can generate genomic, genic and exonic complex rearrangements in humans. *Nature Genetics* **41**, 849-853 (2009).

REVIEWERS' COMMENTS

Reviewer #1 (Remarks to the Author):

Comments for the authors

Review for “DNA transposons mediate duplications via transposition-independent and -dependent mechanisms in metazoans”.

This is a revision. My comments from the first round are well addressed and I think overall the manuscript is in a good shape. Unfortunately, there is something I missed during last round of review and I would like to get it straight.

Row 295 – 299 “Specifically, the protein-coding gene Ssk (Snakeskin, essential for intestinal barrier function), encoded in chromosome 3L (coordinate, 20.2 Mb) was almost completely (5’ upstream together with the genic region except ~200-bp 3’ UTR) captured by FB4, a TcMariner element (Fig. 3A).”
Row 304 – 307 “All three loci of Ssk-FB4 share the same chimeric structure but distinct TSDs (TACATATATG at chr3R: 14.3 Mb, AAATTA AAC at chr3R: 17.7 Mb, and CATGTAGCG at chrX: 2.7 Mb), which suggests recurrent transpositions.”

I am not sure why the authors call FB4 TcMariner (Tc1/Mariner). I also see that this element is called Tc1/Mariner in Repbase and I would guess that was where it came from. However, I don’t see any feature of this element belonging to Tc1/Mariner.

I read some of the literatures cited by the authors. Particularly the work by Badal et al. [Badal, M., Xamena, N. & Cabré, O. FB-NOF is a non-autonomous transposable element, expressed in *Drosophila melanogaster* and present only in the melanogaster group. *Gene* 526, 459-463 (2013).]

It looks like prior to Badal’s work, FB elements were classified as Mutator-like elements (MULEs). I think it makes sense in that FB elements have extended long terminal inverted repeats, 9 to 10 bp target site duplication (TSD), and preferentially inserted at the 5 end of genes. All of those are typical features of MULEs. Particularly, the length of TSD is very diagnostic in terms of which superfamily an element belongs to. Tc1/Mariner elements always have “TA” as their TSD, so there is no way that this element is a Tc1/Mariner element.

Badal’s work only showed that FB-NOF did not encode any transposase, indicating this element is a non-autonomous element. However, most MULEs (as well as most DNA transposons) are non-autonomous element, so I don’t understand why this work represents a piece of evidence to disprove this element is a MULE. In my opinion, it is still fair to call it a MULE, so Ssk-FB4 is basically a Pack-MULE. Certainly, it would be more confident if a FB element containing MULE transposase is identified, but the current evidence is reasonably good.

I have to say the classification of Ssk-FB4 is not super important for this manuscript. Nonetheless, this issue is pretty straight to address so I think the authors should get it corrected. Especially, it looks like the authors may continue to work on this element, so it would be good to get things right from the beginning.

My suggestion is to call it a MULE or a Pack-MULE, and the authors may cite (Feschotte, C., Pritham, E.J., 2007. DNA transposons and the evolution of eukariotic genomes. *Annu. Rev. Genet.* 41, 331–368.) for the classification. If the authors are not comfortable to call it a MULE, they may state that the classification of this element is still controversial so it is an unknown TIR DNA transposon. In my opinion, it is unacceptable or misleading to call it a Tc1/Mariner.

Reviewer #2 (Remarks to the Author):

The authors have done a thorough and excellent job addressing my initial concerns, and went above and beyond to address potential issues with their KO experiments, which is commendable. I have no further concerns, and recommend that the paper should be accepted without additional revisions.

Reviewer #3 (Remarks to the Author):

I'm satisfied with the current version of the manuscript. I think it's ready for publication.

Reviewer #1 (Remarks to the Author):

Comments for the authors

Review for “DNA transposons mediate duplications via transposition-independent and -dependent mechanisms in metazoans”.

This is a revision. My comments from the first round are well addressed and I think overall the manuscript is in a good shape. Unfortunately, there is something I missed during last round of review and I would like to get it straight.

Row 295-299 “Specifically, the protein-coding gene Ssk (Snakeskin, essential for intestinal barrier function), encoded in chromosome 3L (coordinate, 20.2 Mb) was almost completely (5’ upstream together with the genic region except ~200-bp 3’ UTR) captured by FB4, a TcMariner element (Fig. 3A).”

Row 304-307 “All three loci of Ssk-FB4 share the same chimeric structure but distinct TSDs (TACATATATG at chr3R: 14.3 Mb, AAATTAAAC at chr3R: 17.7 Mb, and CATGTAGCG at chrX: 2.7 Mb), which suggests recurrent transpositions.”

I am not sure why the authors call FB4 TcMariner (Tc1/Mariner). I also see that this element is called Tc1/Mariner in Repbase and I would guess that was where it came from. However, I don’t see any feature of this element belonging to Tc1/Mariner.

I read some of the literatures cited by the authors. Particularly the work by Badal et al. [Badal, M., Xamena, N. & Cabre, FB-NOF is a non-autonomous transposable element, expressed in Drosophila melanogaster and present only in the melanogaster group. Gene 526, 459-463 (2013).]

It looks like prior to Badal’s work, FB elements were classified as Mutator-like elements (MULEs). I think it makes sense in that FB elements have extended long terminal inverted repeats, 9 to 10 bp target site duplication (TSD), and preferentially inserted at the 5 end of genes. All of those are typical features of MULEs. Particularly, the length of TSD is very diagnostic in terms of which superfamily an element belongs to. Tc1/Mariner elements always have “TA” as their TSD, so there is no way that this element is a Tc1/Mariner element.

Badal’s work only showed that FB-NOF did not encode any transposase, indicating this element is a non-autonomous element. However, most MULEs (as well as most DNA transposons) are non-autonomous element, so I don’t understand why this work represents a piece of evidence to disprove this element is a MULE. In my opinion, it is still fair to call it a MULE, so Ssk-FB4 is basically a Pack-MULE. Certainly, it would be more confident if a FB element containing MULE transposase is identified, but the current evidence is reasonably good.

I have to say the classification of Ssk-FB4 is not super important for this manuscript. Nonetheless, this issue is pretty straight to address so I think the authors should get it corrected. Especially, it looks like the authors may continue to work on this element, so it would be good to get things right from the beginning.

My suggestion is to call it a MULE or a Pack-MULE, and the authors may cite (Feschotte, C., Pritham, E.J., 2007. DNA transposons and the evolution of eukariotic genomes. *Annu. Rev. Genet.* 41, 331-368.) for the classification. If the authors are not comfortable to call it a MULE, they may state that the classification of this element is still controversial so it is an unknown TIR DNA transposon. In my opinion, it is unacceptable or misleading to call it a Tc1/Mariner.

Many thanks for your professional and insightful comments. We also deeply appreciate your responsibility and patience during the second round of review.

Yes, we previously followed the TE classification of UCSC Genome Browser (Methods, Line 679-684) based on the Repbase library and therefore classified *FB4* as a *TcMariner* TE. Now, as pointed out by the referee, we fully agree that the 9 to 10-bp target site duplications (TSDs) of *FB4* strongly suggest it might not be a *TcMariner* element, since TSDs of *TcMariner* TEs are always 'TA'. However, we'd like to make it clear that the long TIRs of *FB4* are not sufficient to indicate that it belongs to MULE, because a subset of *TcMariner* TEs also have long TIRs¹. For example, *Sleeping Beauty* (SB) has 225-bp TIRs. We further noticed that the subterminal regions of TIRs in *FB4s* are composed of short direct repeats (Fig. R1). Such feature is very similar to that of *Mutal*, which is an active MULE with functional transposase found in the mosquito *Aedes aegypti*², while *TcMariner* TEs (e.g., SB) do not have these direct repeats in TIRs. Together with the 8 to 9-bp TSDs of *Mutal*, *FB4s* very likely belong to MULE rather than *TcMariner* superfamily.

Figure R1. Structure features of *FB4* at chr3L: 20.8 Mb, which mediated the emergence of *Ssk-FB4*. TIRs and non-TIR TE sequences are shown as gray and black boxes. Direct repeats are shown as gray arrows. The length and nucleotides of the direct repeats are variable, and the consensus sequences are shown here.

As the referee pointed out, such a classification issue is not the major focus of this manuscript. So, we revised the text briefly.

“Specifically, the protein-coding gene *Ssk* (*Snakeskin*, essential for intestinal barrier function)³⁸ encoded in chromosome 3L (coordinate, 20.2 Mb) was almost completely (5' upstream together with the genic region except ~200-bp 3' UTR) captured by *FB4*, which was annotated as a *TcMariner* element in Repbase (Fig. 3a).” (Line 305-308)

“Consistently, *FB4* is known as a nonautonomous element transposed by an unknown transposase^{40,41}. Notably, since *FB4* has long TSDs (9 to 10-bp) and TIRs (~700 bp), it

more likely belongs to MULE rather than *TcMariner* superfamily, which generally have short TSDs (2-bp TA) and TIRs (<100 bp)⁵. Thus, *Ssk-FB4s* represent Pack-MULEs in *Drosophila*.” (Line 317-321)

Reviewer #2 (Remarks to the Author):

The authors have done a thorough and excellent job addressing my initial concerns, and went above and beyond to address potential issues with their KO experiments, which is commendable. I have no further concerns, and recommend that the paper should be accepted without additional revisions.

Many thanks for your encouraging comments and we are happy to hear that we have addressed all of your concerns.

Reviewer #3 (Remarks to the Author):

I'm satisfied with the current version of the manuscript. I think it's ready for publication.

Thank you for the encouraging comment.

References

1. Plasterk, R.H.A., Izsvák, Z. & Ivics, Z. Resident aliens: the *Tc1/mariner* superfamily of transposable elements. *Trends in Genetics* 15, 326-332 (1999).
2. Liu, K. & Wessler, S.R. Functional characterization of the active *Mutator*-like transposable element, *Muta1* from the mosquito *Aedes aegypti*. *Mobile DNA* 8, 1 (2017).